# Coupled thermo–geophysical inversion for permafrost monitoring

**Soňa Tomaškovičová and Thomas Ingeman-Nielsen**

Department of Environmental and Resource Engineering, Technical University of Denmark, Nordvej 119, 2800 Kongens Lyngby, Denmark

**Correspondence:** Soňa Tomaškovičová (soto@dtu.dk)

**Abstract.** This study explores an alternative way of deriving soil thermal properties from surface geophysical measurements. We combined ground surface temperature time series with time lapse geoelectrical acquisitions measured from the ground surface in a fully coupled inversion scheme to calibrate a heat conduction model. The quantitative link between the thermal and geoelectrical parts of the modeling framework is the temperature-dependent unfrozen water content, which is also the main factor influencing electrical response of the ground. The apparent resistivity data were incorporated into the coupled framework without being inverted separately, thus reducing the uncertainty inevitably associated with inverted resistivity models. We show that geoelectrical time lapse data are useful as alternative calibration data and can provide as good results as borehole temperature measurements. The fully coupled modeling framework using field data achieved performance comparable to calibration on borehole temperature records in terms of model fit within 0.6 °C, inversion convergence metrics, as well as the predictive performance of the calibrated model.

## 1 Introduction

Numerical modeling is a powerful – and often the only available – tool for assessing the current and forecasting the future thermal state of permafrost (Riseborough et al., 2008; Harris et al., 2009). Models rely on quality data for forcing, calibration and validation. In thermal modeling of permafrost, calibration data are ideally ground temperature time series measured in boreholes, as these provide the most direct information about the ground thermal regime. Boreholes are, however, geographically sparse while providing only discrete information in one spatial dimension. Meanwhile, the ground thermal regime is highly variable due to local conditions.

Surface geophysical measurements offer an attractive way of informing permafrost thermal models. Depending on the geophysical method used, they provide a 2D or 3D picture of subsurface properties and cover comparatively large areas. Repeated measurements have been shown to hold information about in situ processes, guiding the development of more accurate process-based models. Studies by Hoekstra and McNeill (1973), Olhoeft (1975), Scott and Kay (1988), Hauck et al. (2008), Krautblatter et al. (2010), Magnin et al. (2015), Wu et al. (2017), Tomaškovičová (2018), Tang et al. (2018), Holloway and Lewkowicz (2019), Uhlemann et al. (2021), and Tomaškovičová and Ingeman-Nielsen (2023) demonstrate that there is a quantitative link between the electrical and thermal properties of geological materials.

In permafrost thermal modeling with field data, coupling approaches have been applied essentially in two ways: (i) temperature-calibrated resistivity tomography has been used for quantitative estimation of ground ice and water content changes (Krautblatter et al., 2010), and (ii) inverted resistivity models have been used to constrain ground ice change estimates (Hauck et al., 2008). Tomaškovičová et al. (2012) presented a concept of a fully coupled thermo–geophysical inversion, where apparent electrical resistivity data before inversion were used to constrain the optimization of thermal model parameters. Jafarov et al. (2020) demonstrated the feasibility of the approach on synthetic datasets.

In this work, we evaluated the performance of the fully coupled thermo–geophysical optimization framework on field monitoring data. We demonstrated that thermal parameters of a real ground undergoing phase change can be calibrated using time lapse geoelectrical measurements collected from the ground surface. The electrical properties of

the ground depend mainly on the amount of unfrozen water available to carry the current. This unfrozen water content is temperature dependent, and temperature at any depth depends on the surface energy balance (including water bal-
5 ance) and the soil thermal properties. We used electrical resistivity data for calibration because of their comparative ease of acquisition, including the possibility of automation, and relative ease and speed of data processing. However, any kind of geophysical data are in principle suitable, as long
as the petrophysical relationship between ground temperature and the targeted geophysical property can be calibrated. Borehole temperature records are not needed for thermal model calibration in the fully coupled inversion approach; however, where available, they present valuable validation
data. The workflow of the fully coupled thermo–geophysical inversion is explained in the following section.

## 2 The concept of the fully coupled thermo–geophysical inversion

The fully coupled thermo–geophysical inversion approach
aims at predicting ground temperatures using geophysical measurements for the calibration of the thermal parameters of a ground thermal model. When using electrical resistivity data for calibration, the approach is built on the quantitative link between ground temperature and ground electrical
properties (Hauck, 2002; Hauck et al., 2008; Doetsch et al., 2015; Oldenborger and LeBlanc, 2018). The ground electrical properties depend on four main factors: soil mineralogical composition, soil porosity, fraction of unfrozen water content and geochemical composition of the pore water (e.g., Hoek-
stra et al., 1975; Friedman, 2005). The unfrozen water is typically the dominant conducting phase in a soil. It is also the only component that substantially changes its volume fraction over the course of a year due to temperature-dependent processes of freezing and thawing (as well as due to drying
and wetting in the unfrozen part of the year). Due to the zero-curtain effect, there is no single temperature value at which ground resistivity changes from frozen to unfrozen (Hauck, 2002; Doetsch et al., 2015; Tomaškovičová and Ingeman-Nielsen, 2023). It is the unfrozen water content that quan-
titatively links the temporal changes in ground temperature with changes in ground electrical resistivity. The coupled inversion approach is outlined in Fig. 1 and further explained herein.

The coupled model consists of two, essentially standalone,
modules: a heat conduction model (described in Sect. 4) and an electrical resistivity model (described in Sect. 5). The 1D heat conduction model calculates a temperature distribution in the ground given a set of initial and boundary conditions and of initial thermal parameter estimates.
The calculated temperature distribution is then translated into a 1D multi-layer geoelectrical model by weighting the specific resistivities of the ground constituents by their

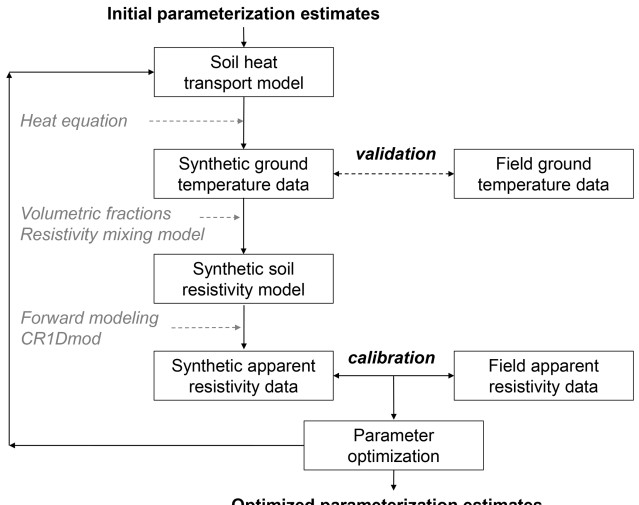

**Figure 1.** Flow diagram of the fully coupled thermo–geophysical inversion using ground surface temperature data as model forcing and apparent resistivity data collected from the ground surface for calibration.

temperature-dependent volumetric fractions. From the geo-electrical model, the forward apparent resistivity response is calculated using the same electrode configurations as on 55 the field site. The calculated apparent resistivities are compared to the field geoelectrical measurements. The difference is then minimized by adjusting the thermal parameters of the heat conduction model, from which the forward resistivities were calculated, and the specific resistivities of the soil con- 60 stituents.

The key characteristic of the fully coupled approach is the use of apparent resistivities for calibration, instead of inverted resistivity models. The reason for using apparent resistivities is the expectation that they introduce less ad- 65 ditional uncertainty to the calibration in the form of inherent inversion assumptions and artifacts. The relationship translating a certain ground electrical composition into apparent resistivity is unique and governed by equations for the conservation of charge, Ohm's law and the geometry of 70 electrode configuration used to collect the resistivity data. Conversely, any inverted resistivity model is only one of a large number of possible realizations that explain the measured apparent resistivity data acceptably well. The interpretation of resistivity measurements from permafrost ter- 75 rain in particular suffers from strong resistivity contrasts and ground ice features that may lead to over-estimating resistivity model parameters (Ingeman-Nielsen, 2005; Supper et al., 2014; Tomaškovičová and Ingeman-Nielsen, 2023). The non-unique nature of the inverted resistivity models thus 80 provides a less-solid basis for quantitative calibration.

We note that, while the conversion from a ground resistivity model to an apparent resistivity response is unique, the number of thermal parameter value combinations able

to explain the measured apparent resistivities remains infinite. Hence, the choice of parameter bounds, optimization method and constraints, as well as careful interpretation of the inverted parameters in terms of their physical plausibility and modeling assumptions, are necessary.

The fully coupled modeling framework does not rely on borehole temperature data for the thermal model calibration; only ground surface temperatures, or possibly downscaled air temperatures, are needed to drive the model. Naturally, when available, borehole temperature records are a very useful resource for model validation and performance evaluation. We use borehole temperatures in our study to (i) validate the heat conduction model formulation and implementation, (ii) validate the inverse problem formulation, and (iii) quantitatively assess the performance of the fully coupled inversion framework by comparing the performance of the thermal model calibrated on resistivities to the performance of the thermal model calibrated on ground temperatures.

## 3 Field site and data

The data used in this work come from an automated permafrost monitoring station in Ilulissat, West Greenland. The Ilulissat monitoring site (69°14′ N, 51°3′ W; 33 m above sea level) is situated ca. 200 m east of the airport in Ilulissat, on the mainland, in the inner part of the Disko Bay. The mean annual air temperature (MAAT) between 2010–2019 was −3.1 °C (data from Cappelen, 2020). According to Obu et al. (2019) and Brown et al. (1998), the site is located in the continuous permafrost zone. Borehole temperature records from the years 2012–2015 from the site (Tomaškovičová and Ingeman-Nielsen, 2023) yield the following permafrost parameters: the maximum active layer thickness is around 0.9 m , the ground temperatures at 4 m depth are in the range of −3.0 to −3.4 °C. The depth of zero annual amplitude is 5 m (defined as the depth of maximum annual amplitude < 0.1 °).

The sedimentary profile in Ilulissat consists of fine-grained marine sediments deposited during the sea transgression following deglaciation of the area some 9600 BP (Bennike and Björck, 2002). The bedrock – Nagsugtoquidian gneisses with amphibolitic bands – is encountered at 7 m depth according to borehole 78020 drilled as part of the site investigations for the nearby airport in 1978 (Geoteknisk Institut, 1978). The soils at the site are classified as silty to very silty clays according to the Danish engineering geological practice (Larsen et al., 1995), which is normally used in Greenland. The soil column is covered by a few centimeter-thick vegetation layer.

Tomaškovičová and Ingeman-Nielsen (2023) showed that the thermal regime in the sediments at the site is strongly influenced by hysteresis of unfrozen water content between freezing and thawing periods. The magnitude of this hysteresis justifies use of separate parameterization for the freezing and thawing periods, respectively.

## 4 Heat conduction model

We set up a ground thermal model based on the one-dimensional heat conduction equation with phase change (Lunardini, 1981):

$$\left(C_e + L\frac{\partial}{\partial T}\theta_w(T, x)\right)\frac{\partial}{\partial t}T(x, t) = \frac{\partial}{\partial x}\lambda_e\frac{\partial}{\partial x}T(x, t). \quad (1)$$

In this formulation, $T$ (°C) is temperature; $L$ ($\mathrm{J\,m^{-3}}$) is the volumetric latent heat of phase change between water and ice; $\theta_w$ is the volumetric unfrozen water content of the bulk soil ($\mathrm{m^3_{water}\,m^{-3}_{bulk}}$); $C_e$ ($\mathrm{J\,m^{-3}\,K^{-1}}$) and $\lambda_e$ ($\mathrm{W\,m^{-1}\,K^{-1}}$) are effective heat capacity and effective thermal conductivity, respectively, of the multi-phase media under consideration; $x$ (m) is the depth below ground surface; and $t$ (s) is the time. Equation (1) applies under the assumptions that there are no additional internal sources or sinks of heat, that no volume change is associated with the phase changes, that migration of water is negligible, and that there are no lateral variations in topography and soil properties (standard 1D assumption).

Dirichlet boundary conditions are applied at the top and bottom of the model (at depths $x = 0$ and $x = l$, respectively), such that $T(0, t) = T_u(t)$ and $T(l, t) = T_l(t)$, where subscripts $u$ and $l$ denote the upper and lower boundaries. A fixed temperature is used as the bottom model boundary, as the yearly temperature amplitude at the bottom of 6 m deep borehole is < 0.09 °C. The initial temperature distribution is specified throughout the model domain, such that $T(x, 0) = T_0(x)$, where $T_0(x)$ is the temperature at depth $x$ m and time $t = 0$ s.

Following Lovell (1957) and Anderson and Tice (1972), we use a power function to describe the soil unfrozen water content variation at temperatures below freezing point:

$$\theta_w = \eta\phi, \quad \phi = \begin{cases} S & T \geq T^* \\ \alpha|T_f - T|^{-\beta} & T < T^*, \end{cases} \quad (2)$$

where $\theta_w$ is the volumetric unfrozen water content of the bulk soil ($\mathrm{m^3_{water}\,m^{-3}_{bulk}}$), $\eta$ is the porosity ($\mathrm{m^3_{voids}\,m^{-3}_{bulk}}$), $\phi$ is the volumetric unfrozen pore water fraction ($\mathrm{m^3_{water}\,m^{-3}_{void}}$), $S$ is the water saturation in a completely unfrozen state ($\mathrm{m^3_{water}\,m^{-3}_{void}}$) (assumed unity in this study), and $\alpha$ and $\beta$ are empirical positive valued constants describing the intrinsic freezing characteristics of the given soil. $T^*$ (°C) is the effective freezing point of the bulk soil – the lowest temperature at which all the water in the soil remains unfrozen ($\phi = S$) – and is given by

$$T^* = T_f - \left(\frac{S}{\alpha}\right)^{-\frac{1}{\beta}}, \quad (3)$$

where $T_f$ [°C] is the freezing point of the pore water as a free substance.

Given a certain value of unfrozen water content, and under the assumption that the soil is fully saturated at all times, the volumetric fractions of soil particles $\theta_s$ and ice $\theta_i$ are derived as

$$\theta_s = 1 - \eta, \quad \theta_i = \begin{cases} 0 & T \geq T^* \\ \eta \, (S - \phi) & T < T^*. \end{cases} \tag{4}$$

Pronounced hysteresis in the unfrozen water content variation (e.g., Pellet and Hauck, 2017; Pellet et al., 2016; Overduin et al., 2006) and its effect on the yearly enthalpy change (Tomaškovičová and Ingeman-Nielsen, 2023) normally require that two separate parameterizations – ($\alpha_f$, $\beta_f$) for the freezing season and ($\alpha_t$, $\beta_t$) for the thawing season in Eq. (2) – are used to accurately model the freezing and thawing processes, respectively. In this modeling exercise, we use data from the freezing seasons only; therefore, our notation ($\alpha$, $\beta$) relates to the freezing curve parameterization (instead of notation ($\alpha_f$, $\beta_f$)).

The effective parameters of a bulk, three-phase soil are derived as a function of their respective volumetric fractions, which are essentially a function of temperature. The effective heat capacity $C_e$ is expressed as the sum of the specific heat capacities of the soil phases weighted by their volumetric fractions (e.g., Anderson et al., 1973):

$$C_e = C_s \theta_s + C_w \theta_w + C_i \theta_i. \tag{5}$$

Common Johansen's thermal parameterization (geometric mean) is used for modeling the effective thermal conductivity $\lambda_e$ of a $n$-phase soil (Johansen, 1977; Zhang et al., 2008):

$$\lambda_e = \prod_{j=1}^{N} \lambda_j^{\theta_j}. \tag{6}$$

To solve the heat conduction equation, we used an in-house code soilfreeze1D, which implements a finite-difference scheme on a fixed grid with equidistant nodes. The code uses the unconditionally stable Crank–Nicholson algorithm with adaptive time stepping to minimize errors in the solution. For sufficiently small time steps, the analytical derivative of Eq. (2) may be used to estimate the change in unfrozen water content. However, to allow manageable step sizes, we implemented an iterative scheme for the change in water content. The first iteration for each time step uses the analytical derivative, while subsequent iterations use a finite difference, based on the temperature estimate resulting from the previous iteration. Iterations proceed until the maximum change in estimated temperature is less than a specified threshold, or until a specified number of iterations have completed, in which case the time step is reduced.

The lithology at the Ilulissat field site justifies that a number of simplifying assumptions are made in the interest of maintaining the model parsimony:

1. Heat conduction is assumed to be the dominant mechanism of heat transport. This is reasonable considering the waterlogged properties of the silty clays at the site with low hydraulic conductivity. A similar assumption was previously successfully used by, e.g., Nicolsky et al. (2007, 2009).

2. The ground is assumed to be fully saturated, thus consisting of up to three constituents: soil particles, water and ice. This is a realistic assumption at our site during the time period used for calibration, as evidenced by unfrozen water content measurements reported in Tomaškovičová and Ingeman-Nielsen (2023).

3. The ground is assumed to be homogeneous, thus neglecting any lithological layers or varying thermal properties. This is true in terms of the soil type, which is uniform throughout the modeled soil column. Heterogeneities, however, are present, namely in the form of ice lenses (mainly in the depth of between 0.9–1.5 m) and increasing pore water salinity (in the depth of below 4 m).

4. The specific heat capacity and specific thermal conductivity of the respective soil constituents are assumed constant, i.e., not dependent on temperature or salinity. This is an acceptable approximation, as using constant parameters resulted in errors of less than 10 % in the calculation of the effective (bulk) thermal properties in the temperature range of between −20 and 0 °C (Osterkamp, 1987).

5. The latent heat of phase change varies with unfrozen water content, but using a constant value has been proved satisfactory for temperatures above −20 °C (Anderson et al., 1973).

6. A fixed temperature is used as the bottom model boundary. This is an acceptable simplification when modeling relatively short temperature time series of 180 d and considering that measured yearly temperature amplitude at the bottom of 6 m deep borehole is < 0.09 °C.

We distinguish principle limitations of the method from simplifying assumptions and further discuss their implications in Sect. 8.

The choice of model discretization in time and space was based on convergence testing. For this modeling experiment, the thermal model domain was set to be 6 m deep. The comparatively shallow model was adequate considering the relatively short time series we modeled – up to 180 d, given by the need for separate parameterizations of freezing and thawing seasons (Sect. 3). We specified an equidistant mesh for the heat conduction model solution with nodes every 0.05 m, and we limited the maximum step size of the differential equation solver to 1 h. The forcing data for the model were ground surface temperatures collected every 3 h. The fixed

temperature at the model bottom boundary was informed by the observations from the 6 m deep borehole, located ca. 10 m away from the resistivity acquisition line.

The modeling and inversion frameworks were implemented in Matlab, with the heat equation solver soilfreeze1D implemented in Python using the NumPy module for optimized array and matrix computations.

## 5 Resistivity model

The geophysical part of the modeling framework consists of a 1D forward geoelectrical model. Through convergence testing, we determined that 128 layers of equal thickness produced a convergent solution for the 6 m deep model. A representative temperature was assigned to each resistivity model layer by interpolating the nearest heat conduction model solution. Fractions of water, ice and soil minerals in each layer were calculated based on Eqs. (2) and (4). The effective bulk resistivity $\rho_e$ of each resistivity model layer was then derived using a resistivity–mixing relationship. We considered two commonly used resistivity–mixing relationships: (i) Archie's law and (ii) the geometric mean model. The choice between them was made based on their parameters' sensitivities to calibration in the fully coupled inversion scheme (Fig. 7).

The traditional Archie's law (Archie, 1942) derives effective bulk resistivity of ground material based on the material's porosity and on the resistivity of the pore fluid:

$$\rho_e = \rho_w \eta^{-m} \phi^{-n}, \tag{7}$$

where $\rho_e$ is the effective resistivity of the bulk soil, $\rho_w$ is the specific resistivity of the pore water, $\eta$ is the porosity, $\phi$ is the unfrozen fraction of pore water, and $m$ and $n$ are empirical coefficients.

The geometric mean model (e.g., Guéguen and Palciauskas, 1994) estimates the effective bulk resistivity as the geometric mean of specific resistivities of the respective ground components, weighted by their volumetric fractions:

$$\rho_e = \left( \prod_{i=1}^{n} \rho_i^{\theta_i} \right), \tag{8}$$

where $\rho_i$ and $\theta_i$ are the specific resistivity and volumetric content of the $i$th soil constituent, respectively.

Based on the derived effective ground resistivity model, synthetic apparent resistivities $\rho_s$ were forward-calculated using CR1Dmod code (Ingeman-Nielsen and Baumgartner, 2006), using the same electrode configuration as in the field acquisitions. As the apparent resistivity field measurements launched every day at 18:00 UTC and last for up to 5.5 h, the thermal model solutions at time steps between 18:00–00:00 UTC every day were averaged to provide the temperature profile corresponding to the timing of the resistivity acquisitions.

## 6 Validation of the heat conduction model and of the inverse problem formulation

Prior to utilizing the heat conduction model as part of the coupled inversion, we evaluate its ability to reproduce real ground temperature dynamics. We also evaluate the performance of the optimization algorithm and identify a strategy for effective multi-parameter optimization.

### 6.1 Choice of the target parameters for optimization

Approaching an inverse problem begins with identifying a subset of model parameters to be targeted by the optimization, as optimization with respect to all model parameters is not feasible. Diagnostic statistics can measure the amount of information for parameter calibration in the available data and can inform the choice of parameters that can be effectively recovered from the data (Hill, 1998).

Scaled sensitivities (SSs) (Hill, 1998) compare the importance of different observations to the estimation of a single parameter (or the importance of different parameters to the calculation of a simulated value). SSs can help to identify the time periods in calibration data with the most information for the calibration of the target parameter. SSs are calculated as

$$ss_{ij} = \left( \frac{\delta y_i'}{\delta v_j} \right) v_j \sqrt{\omega_i}, \tag{9}$$

where $y_i'$ is a simulated value, $v_j$ is the $j$th estimated parameter, $\left( \frac{\delta y_i'}{\delta v_j} \right)$ is the sensitivity of the simulated value to the $j$th parameter and is evaluated at $V$, $V$ is the vector which contains the parameter values at which sensitivities are evaluated, and $\omega_i$ is the weight of the $i$th observation.

To evaluate the relative importance of the respective model parameters for the model predictions, we calculate the composite scaled sensitivities (CSSs) (Hill, 1998) for each of the parameters of the heat conduction model. We chose to demonstrate the thermal model's sensitivity to all parameters as per the heat conduction, Eq. (1), including the ones that are not calibrated in practice (constants). This is because such constants would normally be excluded from a sensitivity analysis while they are, strictly speaking, temperature and salinity dependent (specific heat capacity and thermal conductivity of ice and pore water). The assumption of a constant is a standard simplifying modeling assumption in a study not focusing on the temperature and salinity dependence of these thermal parameters and designed following the principle of model parsimony. Nevertheless, a known temperature and salinity dependence could easily be incorporated into the current model formulation if the model application required it and justified the increased complexity.

Figure 2 shows the change in simulated temperature field caused by a 10 % increase in each of the evaluated parameters. Because the calibration problem is nonlinear with respect to many parameters of interest, the sensitivity of the

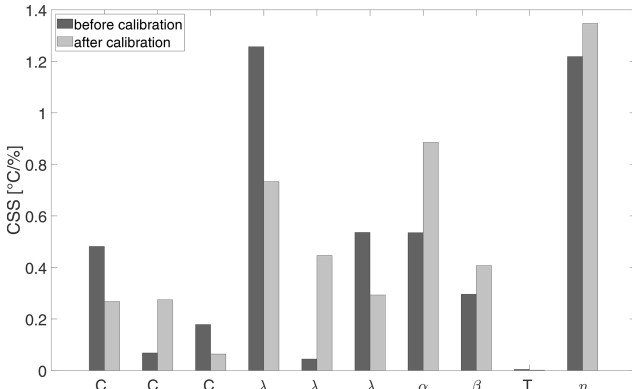

**Figure 2.** Composite scaled sensitivities (CSSs) of 10 parameters of the three-phase heat conduction model, before and after calibration. CSS corresponds to the change in simulated temperature field caused by a 10 % increase in the evaluated parameter. The parameters are $C_s$, $C_w$ and $C_i$ – the heat capacities of soil grains, water and ice, respectively; $\lambda_s$, $\lambda_w$ and $\lambda_i$ – the thermal conductivities of soil grains, water and ice, respectively; $\alpha$ and $\beta$ – the soil freezing parameters; $T_f$ – the freezing point of water as a free substance; and $\eta$ – porosity.

evaluated parameter will change for different values of parameters in the parameter vector $V$, as well as for different model discretizations in time and space. An exhaustive sensitivity analysis at every inversion iteration would, however, be computationally inefficient. Assuming some degree of linearity in the model response to each parameter input, a carefully chosen starting point will be descriptive for the sensitivity of each of the fitted parameters. Less influential parameters, as well as parameters with well-known table values, were fixed during the actual optimization. The most sensitive parameters were then calibrated in a multi-parameter optimization. Please refer to Table 1 for an overview of the fixed and fitted parameters.

According to Fig. 2, before calibration, parameters $C_w$ (heat capacity of water), $C_i$ (heat capacity of ice), $\lambda_w$ (thermal conductivity of water), $\lambda_i$ (ice) and $T_f$ had relatively less influence on the heat conduction model calculations and thus were fixed to table values or an empirical small value ($T_f$). Parameters $C_s$ and $\lambda_s$ (heat capacity and thermal conductivity of mineral grains), $\alpha$ and $\beta$ (freezing and thawing parameters), and porosity ($\eta$) are soil-specific properties with substantial influence on model predictions. After calibration, porosity remained the key parameter for model predictions. The importance of $\alpha$ and $\beta$ parameters slightly increased, while the influence of the heat capacity and thermal conductivity of the mineral phase has relatively decreased.

Figure 3 shows the scaled sensitivity for each of the five fitted parameters of the heat conduction model. As expected, the periods of time when the ground undergoes a phase change contained the most information for the estimation of each given parameter. Ground temperature cooling from

around $-2\,°C$ and lower appears to be the most important part of data acquisition for the model calibration. This could be, e.g., due to ice formation that has progressed enough to alter the bulk thermal properties of the ground as suggested by the highest sensitivity of porosity $\eta$ to this part of the dataset.

### 6.2 Inverse problem formulation

To describe the inverse problem we adopt the notation from (Nicolsky et al., 2007) and (Nicolsky et al., 2009). We define the vector $V$ as consisting of the fitted thermal parameters: $V = \{C_s, \lambda_s, \alpha, \beta, \eta\}$. For each physically realistic control vector $V$, it is possible to compute the temperature dynamics and compare them to measured borehole data. The measured data are organized in a vector $d$ in the form of averages of eight daily temperature records at each sensor depth. The data $d$ are related to the control vector $V$ by $m(V) - d = e$, where $m$ is the modeled counterpart of the data, and $e$ is the misfit vector. In theory, if there are no measurement errors or model inadequacies, the misfit vector $e$ can be reduced to zero. If there are errors in the data or in the model, the aim is to minimize $e$ by varying the control vector $V$.

The optimization of thermal parameters uses the trust-region reflective algorithm based on the interior-reflective Newton method (Coleman and Li, 1996), as implemented in the Matlab solver *lsqnonlin*. The cost function is the root mean square error (RMSE) between measured and simulated ground temperatures. Convergence is identified by meeting at least one of the two criteria: either (1) the change of parameter value or (2) the change in the RMSE between previous and current iterations should be smaller than a prescribed tolerance.

Specifying *bounds* for the fitted parameters spares the solver from examining physically implausible parameter values. An overview of the fixed parameter values and fitted parameter bounds is provided in Table 1. The values of the fixed $C$ and $\lambda$ parameters were chosen based on the most commonly encountered values in the literature. The $T_f$ value was set to be negative but very small. The bounds for the fitted parameters were selected as the widest physically possible bounds (in the case of porosity $\eta$) or the extreme values encountered in the literature. The bounds could be narrowed down to further constrain the optimization based on knowledge of the site conditions.

To identify a viable combination of fitted parameters that can be calibrated simultaneously, we first test the calibration approach in two synthetic scenarios, with and without noise (Sect. 6.3), before applying it to the field data (Sect. 6.4).

### 6.3 Heat conduction model validation of synthetic data

Heat conduction model parameters in the control vector $V$ were set to values drawn from within the limits of Table 2.

**Table 1.** Parameterization of the heat conduction model. For the *fixed parameters*, we show their fixed values in the optimization. For the *fitted parameters*, we list their bounds – maximum and minimum values that the optimization algorithm is permitted to investigate when searching for the optimal parameter value.

| Fixed parameters | Value | Fitted parameters | Bounds |
|---|---|---|---|
| $C_w$ | $4.19 \times 10^6 \, \mathrm{J\,m^{-3}\,K^{-1}}$ | $C_s$ | $0.6 \times 10^6 – 4.1 \times 10^6 \, \mathrm{J\,m^{-3}\,K^{-1}}$ |
| $C_i$ | $1.9228 \times 10^6 \, \mathrm{J\,m^{-3}\,K^{-1}}$ | $\lambda_s$ | $0.5 – 8 \, \mathrm{W\,m^{-1}\,K^{-1}}$ |
| $\lambda_w$ | $0.56 \, \mathrm{W\,m^{-1}\,K^{-1}}$ | $\eta$ | $0.1 – 0.9$ |
| $\lambda_i$ | $2.18 \, \mathrm{W\,m^{-1}\,K^{-1}}$ | $\alpha$ | $0.1 – 5$ |
| $T_f$ | $-0.0001 \, °\mathrm{C}$ | $\beta$ | $0.01 – 5$ |

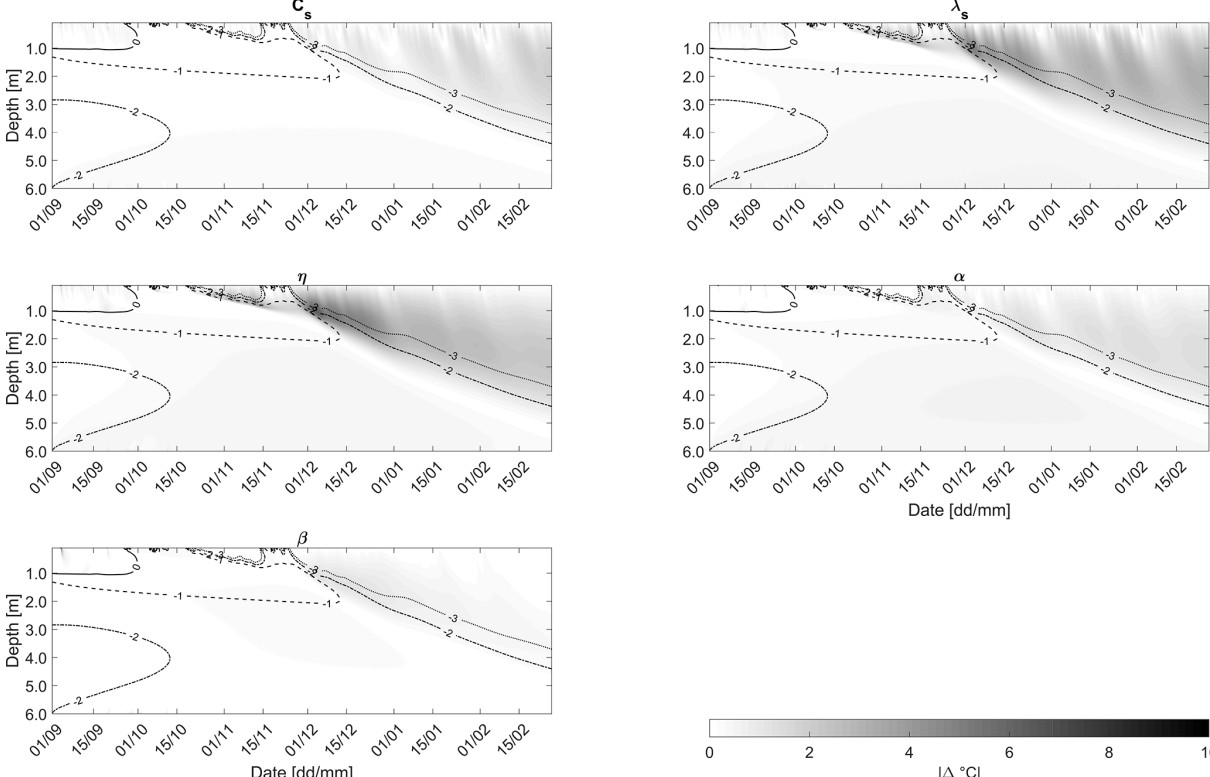

**Figure 3.** Scaled sensitivities of the five fitted parameters of the three-phase heat conduction model. Intensity of the grey shading corresponds to the absolute value of change in the simulated temperature value caused by a 10 % increase in the evaluated parameter. Contour lines with various dashing correspond to isotherms of the ground freeze-up period.

These values were considered the *true parameter values* ($V_t$) and were used to calculate a temperature field $m(V_t)$ in the forward problem (Eq. 1); we termed this temperature field the *reference temperature field*. Next, the control vector $V_t$ was perturbed with an error coefficient $e = [0.5, 1]$ so that the $V_p = V_t \pm e \times V_t$. The resulting temperature field calculated was called the *perturbed temperature field*. We then minimized the RMSE between the perturbed and the reference temperature field by optimizing for one to five of the perturbed fitted parameters at once. The heat conduction model solutions at mesh nodes every 0.1 m between 0.1–1.5 m depth were used in the objective function. This corresponds to the real-life situation, where temperature measurements are typically available from only a limited portion of the soil profile. To force both the reference and perturbed temperature field calculations, we used ground surface temperatures available from a rigid thermistor string at this site (referred to as the "MRC stick" in Tomaškovičová and Ingeman-Nielsen, 2023). The bottom boundary at 6 m was set to $-3.1$ °C based on borehole information from the area.

Each one of the fitted parameters of the heat conduction model ($C_s, \lambda_s, \alpha, \beta, \eta$) converged towards its *true value* in the single-parameter optimization on synthetic data without noise. For porosity ($\eta$), the most sensitive parameter of the

model, when starting the single-parameter calibration from anywhere within the parameter bounds (0.1–0.9) the optimization converged to the true value (0.3) within seven iterations, with RMSE in the order of $10^{-4}$ °C. $\beta$ was the least sensitive of the fitted parameters; nevertheless, the optimization converged to its true value within four iterations, with RMSE in the order of $10^{-4}$ to $10^{-3}$ °C depending on the initial guess. All the parameters were well determined with narrow 95 % confidence intervals.

Up to four parameters could be estimated at once. The joint optimization for the four parameters $\lambda_s$, $\alpha$, $\beta$ and $\eta$ converged within 26 iterations. The recovered parameters were found within 15 % of their true values.

In the next experiment, we added random *noise* with amplitude ±0.03 °C to each of the "measurements" of the reference temperature field. The perturbed parameters were then recovered by optimizing on this "noisy" reference temperature field. True values of all the fitted parameters were recovered in a single-parameter calibration starting from an initial guess of up to 50 % higher than the true parameter value. Equally, the performance of the four-parameter calibration with noise was comparable to the case without noise – recovered parameters laid within 15 % from their true values.

Attempting joint calibration of all five fitted heat conduction model parameters $C_s$, $\lambda_s$, $\alpha$, $\beta$ and $\eta$ caused the optimization to converge towards a solution relatively far from the true parameter values. This approach was therefore deemed not viable. A way to get around fitting the fifth parameter $C_s$ was to define a plausible range for the $C_s$ values and then run a sequence of four-parameter optimizations, with $C_s$ fixed at every step of the predefined range. As a result, we obtained a RMSE value for four-parameter optimizations with $C_s$ fixed at the respective values of the predefined range (such as in Fig. 4a that shows such results for the field, instead of the synthetic, dataset). We call this the *4+1 optimization approach*, and we use it in the next validation step: validation of the heat conduction model on field borehole temperature measurements (Sect. 6.4).

The calibration tests on synthetic datasets confirmed that the trust-region reflective algorithm can recover true parameter values even in scenarios with a reduced number of calibration data, with noisy calibration data, and which are not affected by the use of daily averages instead of individual temperature records. We thus deem the inversion algorithm well suited for handling our optimization problem, provided that the right optimization settings are used. The essential settings for optimal performance of the optimization are the convergence tolerances, size of finite-difference steps, and upper and lower bounds of the permitted parameter value range.

## 6.4    Heat conduction model validation of field data

The next step following the synthetic tests was to confirm that our thermal model succeeds at recovering the thermal parameters of a real ground. This meant to optimize the thermal parameters on borehole temperature time series instead of the synthetic reference temperature field.

Initial and boundary conditions were identical as in the synthetic tests (Sect. 6.3), as these came from a sensor at the site. The difference was that the reference temperature fields in this case were the actual in situ ground temperature time series measured by the rigid thermistor string in the depth of between 0.1–1.5 m during the freezing season of 1 September 2014–28 February 2015 (for description of the sensors and datasets refer to Tomaškovičová and Ingeman-Nielsen, 2023). The reason for choosing the shallow rigid thermistor string records was that they provided the longest uninterrupted series of boundary conditions for forcing our model.

We used the 4+1 optimization approach developed in the synthetic tests (Sect. 6.3). As a maximum of four parameters could be calibrated at once, we began by defining a plausible range for the fifth parameter $C_s$ as $0.6 \times 10^6$–$4.1 \times 10^6$ J m$^{-3}$ K$^{-1}$, subsequently narrowed down to $2.4 \times 10^6$–$3.7 \times 10^6$ J m$^{-3}$ K$^{-1}$. We then ran a total of 92 four-parameter optimizations of the remaining fitted parameters $\alpha$, $\beta$, $\eta$, and $\lambda_s$, with $C_s$ fixed at $0.1 \times 10^6$ increments of the predefined range. The summary of all the optimization runs is provided in Table 2, including the "initial" and the "optimized" parameter values, the average error after optimization ("minimum RMSE"), and the confidence intervals.

The smallest average error (RMSE5 = 0.5503) between the field and simulated temperature fields was found for the following parameter combination: $C_s = 2 \times 10^7$ TS1 J m$^{-3}$ K$^{-1}$, $\alpha = 0.75$, $\beta = 0.10$, $\eta = 0.50$ and $\lambda_s = 1.51$ W m$^{-1}$ K$^{-1}$ (Table 2). This optimization run took 14 iterations to converge. The fit between the field measurements and the temperature field calculated with these parameter values is shown on Fig. 4b. It shows consistently good agreement of the simulations with observations, particularly in the portion of the dataset below the freezing point. The highest misfit is associated with temperatures above the freezing point – when the ground temperature is not largely controlled by phase change processes and water movement and evaporation of the soil moisture may potentially influence the bulk thermal properties.

It is, however, important to point out that the changes in RMSE between all the 92 optimization runs were very small – below the precision of our temperature sensor – in spite of the large range of $C_s$ values evaluated (Fig. 4a). This suggests two conclusions: first, that our implementation of the heat conduction model is not sensitive enough to the parameter $C_s$ to enable its calibration. Second, the optimized model parameter values depend on the initial parameter values of the starting model, which is reflected in the spread of the optimized parameter values producing practically the same model fit. We can therefore expect that the optimized parameter values are not the true ground thermal properties; however, they reflect them well enough to simulate the temperature field within 0.06 °C of the field measurements, on average.

**Table 2.** Summary of the 92 calibration runs on field data using the 4+1 optimization approach. The "RMSE1" shows the minimum RMSE from 36 optimizations runs, each starting with $C_s$ fixed at 0.1 MJ increments between 0.6– 4.1 MJ m$^{-3}$ K$^{-1}$ and remaining parameters $(\alpha, \beta, \eta, \lambda_s)$ starting from the initial values. The RMSE2 through RMSE5 show the minimum RMSE from 14 calibration runs each, starting with $C_s$ fixed at 0.1 MJ increments between 2.4–3.7 MJ m$^{-3}$ K$^{-1}$ (narrower $C_s$ range) and the remaining four fitted parameters starting from the initial values as specified. The 95 % confidence intervals (95 % CI) indicate that the range of values that one can be 95 % certain contains the true mean value of the parameter.

| Run | Parameter | Initial | Optimized | 95 % CI | Minimum RMSE [°C] |
|---|---|---|---|---|---|
| RMSE1 | $C_s$ | 0.6–4.1 | 3 | – | 0.5517 |
| | $\alpha$ | 0.21 | 0.7468 | ±0.0064 | |
| | $\beta$ | 0.60 | 0.1006 | ±0.0089 | |
| | $\eta$ | 0.30 | 0.5258 | ±0.0198 | |
| | $\lambda_s$ | 2.00 | 1.7136 | ±0.0955 | |
| RMSE2 | $C_s$ | 2.4–3.7 | 2.9 | – | 0.5515 |
| | $\alpha$ | 0.10 | 0.7738 | ±0.0053 | |
| | $\beta$ | 0.20 | 0.0839 | ±0.0070 | |
| | $\eta$ | 0.40 | 0.5935 | ±0.0172 | |
| | $\lambda_s$ | 3.00 | 2.0573 | ±0.1328 | |
| RMSE3 | $C_s$ | 2.4–3.7 | 2.4 | – | 0.5509 |
| | $\alpha$ | 0.40 | 0.7916 | ±0.0040 | |
| | $\beta$ | 0.55 | 0.0792 | ±0.0059 | |
| | $\eta$ | 0.35 | 0.6308 | ±0.0145 | |
| | $\lambda_s$ | 1.80 | 2.2407 | ±0.1413 | |
| RMSE4 | $C_s$ | 2.4–3.7 | 2.5 | – | 0.5523 |
| | $\alpha$ | 0.32 | 0.8129 | ±0.0033 | |
| | $\beta$ | 0.70 | 0.0719 | ±0.0049 | |
| | $\eta$ | 0.60 | 0.7276 | ±0.0113 | |
| | $\lambda_s$ | 2.20 | 4.1891 | ±0.3923 | |
| RMSE5 | $C_s$ | 2.4–3.7 | 2.7 | – | 0.5503 |
| | $\alpha$ | 0.50 | 0.7482 | ±0.0061 | |
| | $\beta$ | 0.58 | 0.1045 | ±0.0090 | |
| | $\eta$ | 0.20 | 0.5012 | ±0.0188 | |
| | $\lambda_s$ | 1.90 | 1.5080 | ±0.0713 | |

Figure 5 shows the analysis of differences between the field-measured vs. simulated (using RMSE5 parameterization) temperatures at five depths throughout the active layer (0–0.9 m) and top of the permafrost (below 0.9 m). Predictably, the model struggles to accurately reproduce amplitudes of rapid temperature fluctuations in the shallow subsurface and introduces smoothing and lag into the simulated temperature time series. Overall, above the freezing point, the model introduces a warm bias, with simulated temperatures being warmer than field measurements; conversely, below the freezing point, the model exhibits a cold bias. These observations suggest that the thermal conductivity $\lambda_s$ in the model may be over-estimated compared to the properties of the real ground. Permafrost temperatures (below 0.9 m) are reproduced more accurately, possibly because the ground spends more time in the phase transition around 0 °C, which the model reproduces very well. This suggests that the soil freezing curve parameters $(\alpha, \beta)$ are close to the real soil values.

Since the sensitivity of a model to input parameters changes with the changing values of these parameters, we repeated the sensitivity analysis from Sect. 6.1 using the calibrated parameter values (RMSE5). The CSS analysis confirmed the importance of the fitted parameters $C_s, \alpha, \beta, \eta$ and $\lambda_s$ for the thermal model predictions and did not reveal new optimization targets (Fig. 2, after calibration). We accepted the model calibrated in the RMSE5 optimization run as a well-performing model with physically plausible parameter values, and we proceeded to validate these in the following Sect. 6.5.

## 6.5 Validation of the optimized heat conduction model parameter values

Direct measurements of the heat conduction model parameters were not available, except for porosity $\eta$, which was found to be between 0.40–0.62 depending on the exact location near the site, and depth (Pedersen, 2013). Thus, we chose to validate the model calibration by data splitting (Power,

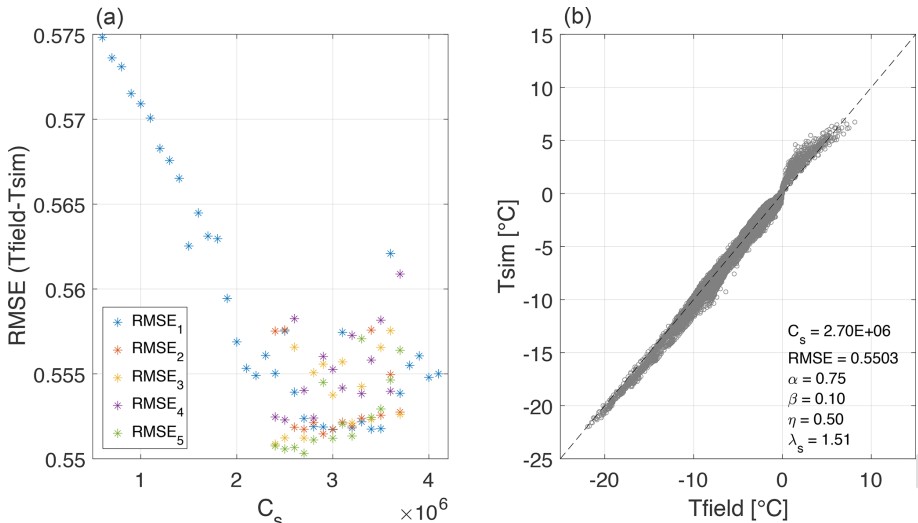

**Figure 4. (a)** RMSE (°C) after optimization for the 92 model runs starting from different initial parameter estimates. The same-color markers indicate the calibrations starting from the same initial values for parameters $\alpha$, $\beta$, $\eta$ and $\lambda_s$ (as specified in Table 2), and the initial value for $C_s$ is fixed on $0.1 \times 10^6$ increments in the specified range ($0.6 \times 10^6$–$4.1 \times 10^6$ J m$^{-3}$ K$^{-1}$ for the group of runs RMSE1 and $2.4 \times 10^6$–$3.7 \times 10^6$ J m$^{-3}$ K$^{-1}$ for RMSE2–RMSE5). **(b)** Cross-plot of field-measured temperature field vs. the temperature field simulated with parameters optimized in the RMSE5 run; the optimized parameter estimates are indicated in the annotation. The average misfit between simulated and field temperatures is 0.55 °C.

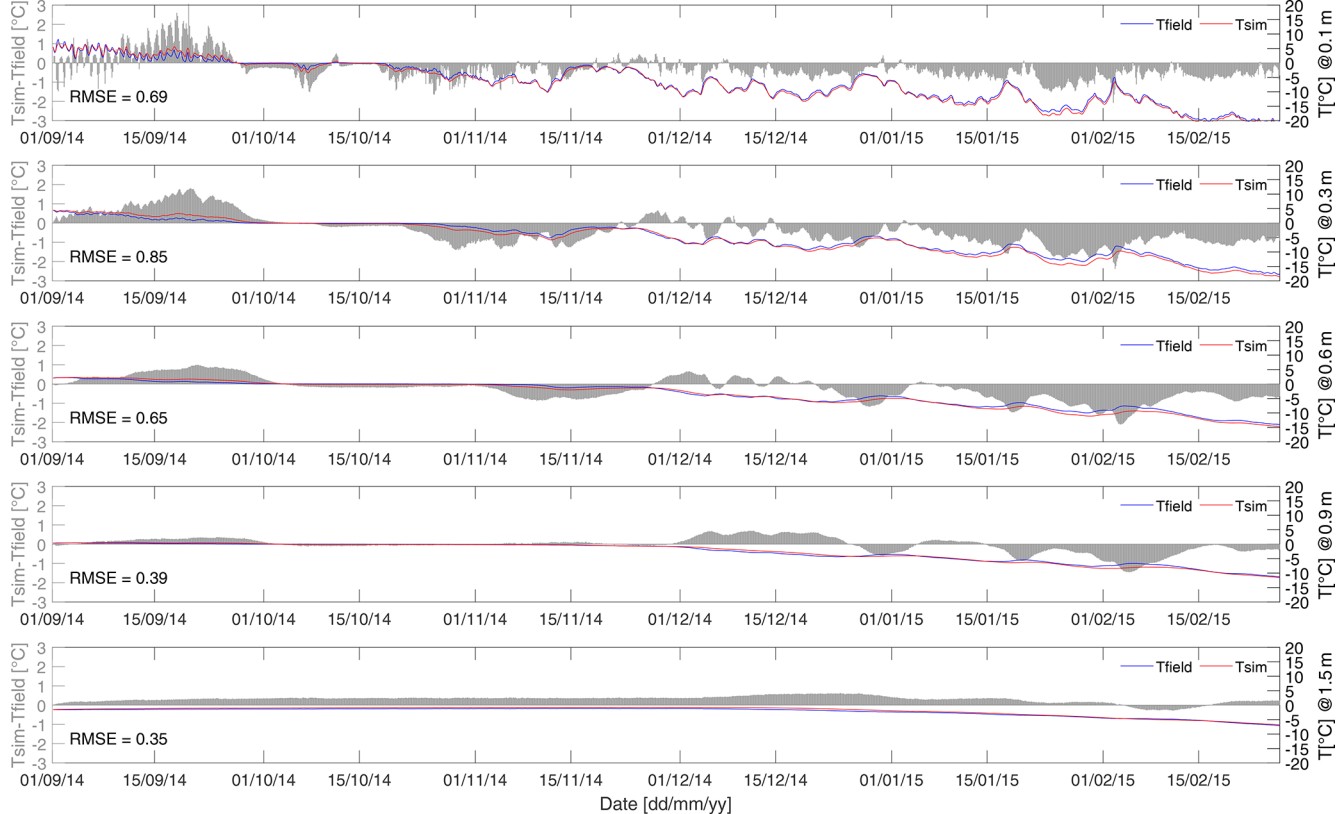

**Figure 5.** Performance of the thermal model calibrated on borehole temperatures evaluated shown as the differences between simulated vs. measured ground temperature time series at five different depths across the active layer and the top of permafrost (below 0.9 m) during the freezing season of 2014/2015. RMSE is in degrees Celsius (°C).

1993). This was done by using the parameter values optimized on the freezing season of 2014/2015 to predict temperature regimes in the previous freezing seasons of 2012/2013 and 2013/2014, respectively.

The model calibrated on the freezing season of 2014/2015 (RMSE5 parameterization) predicted ground temperatures measured between 1 November 2012–28 February 2013 within $\pm 0.63\,^{\circ}\mathrm{C}$ (Fig. 6a). The measurements from the freezing season of 2013/2014 (only 2 months between 1 September 2013–29 October 2013 available) were reproduced within $\pm 0.32\,^{\circ}\mathrm{C}$ (Fig. 6b).

We consider the results of the RMSE5 optimization run to be our best approximation of the actual ground thermal properties in Ilulissat, and we refer to these values when evaluating the performance of other inversion approaches.

## 7 The fully coupled thermo–geophysical inversion

We describe the choice of petrophysical relationship for the resistivity model and then validate the coupled optimization approach on synthetic data before evaluating performance of the fully coupled inversion scheme in comparison to calibration of borehole temperatures.

### 7.1 Choice of the resistivity model parameterization

The sensitivity analysis was repeated for the fully coupled optimization scheme to identify the more suitable petrophysical formulation for use in the resistivity model (Sect. 5). Both thermal and resistivity parameters were evaluated, as the success of the coupled optimization depends on sensitivity of the forward apparent resistivity calculation to change in the heat parameters ($C$, $\lambda$) as well as the resistivity parameters ($\rho$ of the water, ice and soil minerals, Archie's parameters $m$ and $n$). Parameters $\alpha$, $\beta$, $T_{\mathrm{f}}$ and $\eta$ describe the unfrozen water content variation with temperatures below freezing point and are common to both the heat and resistivity modules in the coupled scheme. Figure 7 shows changes in simulated apparent resistivity (log-transformed) resulting from a 10 % change in each of the input parameters for the two petrophysical relationships evaluated – the geometric mean model and Archie's law.

Both petrophysical relationships showed relatively little sensitivity to changes in the thermal model-only parameters $C$ and $\lambda$. Porosity ($\eta$) was among the most influential parameters in the coupled scheme, as it was previously in the thermal model alone. This makes sense, as the total volume (in addition to interconnectedness) of pores available for the storage and movement of soil moisture determines the resistance to current flow. In terms of the resistivity parameters ($\rho_{\mathrm{w}}$ for Archie's law and the geometric mean, $\rho_i$ and $\rho_s$ for the geometric mean only) and the parameters of the soil freezing curve $\alpha$ and $\beta$, Archie's formulation was more sensitive and therefore offered better chances at model calibra-

tion. We therefore proceeded with using Archie's law as the petrophysical relationship in the resistivity module. This was in spite of the two extra parameters to calibrate in Archie's law ($m$ and $n$). Following these considerations as well as the experience from thermal model calibration, we identified our target parameters for optimization as the following: $\lambda_{\mathrm{s}}$, $\alpha$, $\beta$, $\eta$, $\rho_{\mathrm{w}}$, $m$ and $n$.

Parameter optimization in the coupled inversion scheme is based on the same approach as in the thermal modeling alone: the iterative non-linear least-squares formulation using the trust-region reflective algorithm. The cost function is the RMSE between logarithms of field-measured and forward-calculated apparent resistivities. We use log-transformed resistivities, as this way the optimization problem becomes more linear and a more equally weighted fitting of the resistivity data is achieved. The cost function is minimized by adjusting the thermal parameters of the heat conduction model from which the forward resistivities are calculated, as well as the parameters of the resistivity model.

### 7.2 Validation of synthetic data

We performed several optimization runs on synthetic data without noise to get a feel for the sensitivity of the optimization algorithm, correct the optimization settings and identify the combination of parameters that can be estimated at once. In the procedure common with the thermal model testing (Sect. 6.3), a set of arbitrary though realistic parameter values (termed true parameters) was used to produce a reference temperature field, which in turn produced the corresponding *synthetic effective resistivity model* of the ground. From this synthetic resistivity model, reference apparent resistivity response was calculated. The true parameter value(s) were then perturbed by a random error coefficient ranging from $\pm 20\,\%$–90 %. We then aimed to recover the true parameter values by updating the perturbed heat and resistivity parameters iteratively, and comparing the forward-calculated synthetic apparent resistivity to the reference apparent resistivity. The optimization algorithm and the convergence criteria were the same as described in the analogous section on the thermal model testing (Sect. 6.3).

Porosity $\eta$ as the most sensitive parameter of the coupled model was recovered accurately within three iterations and with narrow 95 % confidence intervals. The thermal conductivity of soil grains $\lambda_{\mathrm{s}}$ is an essential parameter for heat conduction model predictions, even though the coupled scheme appeared to have very little sensitivity to it. Nevertheless, in a single-parameter optimization on synthetic data without noise, the true parameter value was recovered accurately within four iterations. Joint calibration of all of the seven fitted parameters at once – $\alpha$, $\beta$, $\eta$, $\lambda_{\mathrm{s}}$, $\rho_{\mathrm{w}}$, $m$ and $n$ – converged within four to five iterations, depending on the starting values, while producing a very good fit between the simulated and reference apparent resistivities (final RMSE after optimization in the range of $10^{-3}$–$10^{-2}$). However, in spite of

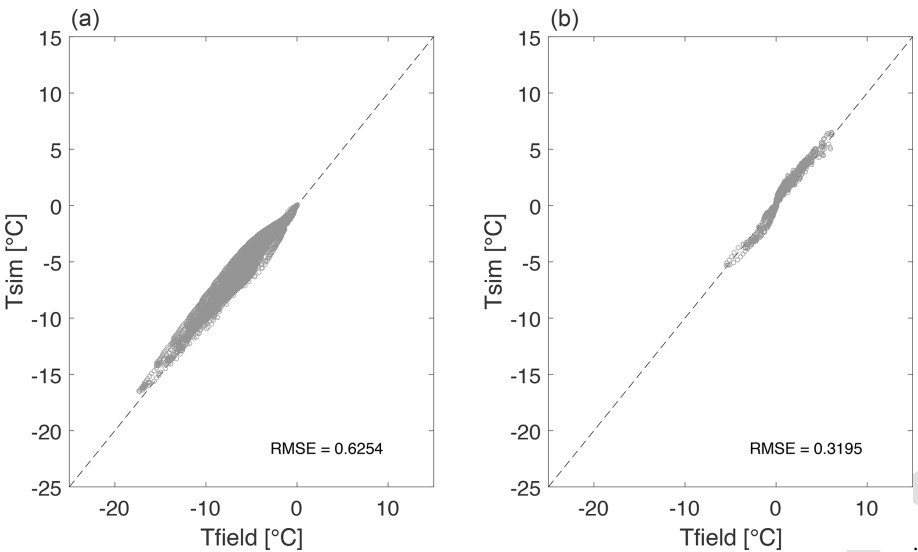

**Figure 6. (a)** Validation of the thermal model calibration of the freezing season of 2012/2013, data between 1 November 2012–28 February 2013. **(b)** Validation of the freezing season of 2013/2014, data between 1 September 2013–29 October 2013. RMSE is in degrees Celsius (°C)

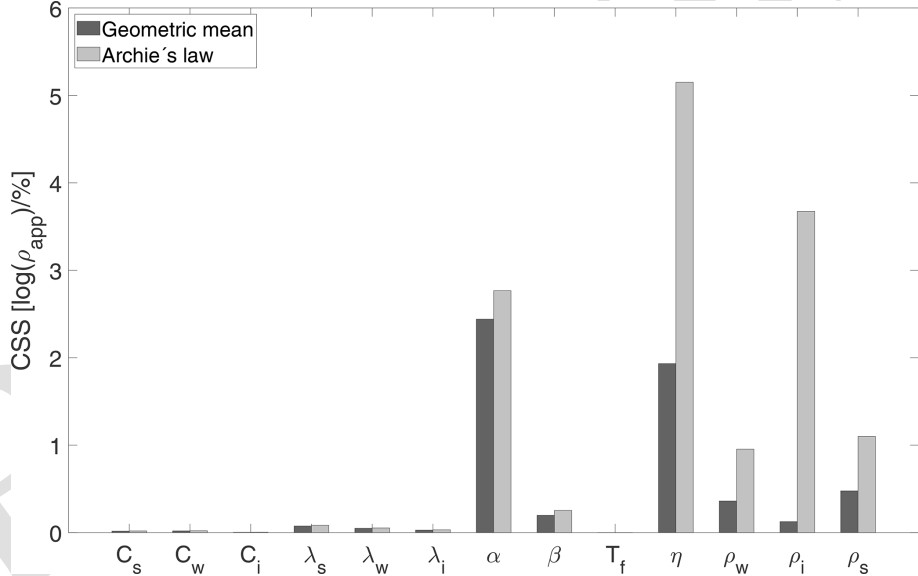

**Figure 7.** Composite scaled sensitivities (CSS) of the 13 parameters of the heat and the resistivity modules of the fully coupled inversion scheme. The sensitivity is expressed as the change in logarithm of forward-calculated apparent resistivity following a 10 % change in the evaluated parameter.

the good model fit, the true parameter values were not accurately recovered, and their optimized values depended on their starting values. In spite of the non-unique results of the inversion, all the optimized parameters lay in physically plausible range.

To improve the accuracy of recovery of the true parameters in the synthetic scenario, we experimented with fixing some of the less sensitive parameters of the coupled scheme. The optimization with six fitted parameters ($\alpha$, $\beta$, $\eta$, $\rho_w$, $m$, $n$)

and $C_s$ and $\lambda_s$ fixed produced a slightly lower final RMSE in comparison to the optimization with seven and five fitted parameters; however, it did not improve the recovery of the true parameter values. We noted that the coupled inversion scheme may not arrive at a unique estimation of the accurate parameter values; nevertheless, our synthetic tests confirmed that the optimization algorithm repeatedly converged to a set of physically plausible parameters while producing a very good fit with the reference dataset.

In the following section, we applied the fully coupled inversion approach to the recovery of thermal and resistivity parameters of the real ground, and we compared the results to the traditional method of calibration of borehole temperatures only.

## 7.3  Fully coupled inversion with field data

Following the experience from synthetic testing (Sect. 7.2), we chose to optimize six fitted parameters at once: $\alpha$, $\beta$, $\eta$, $\rho_w$, $m$ and $n$. We fixed the thermal conductivity $\lambda_s$ to the value $1.70\,\text{W}\,\text{m}^{-1}\,\text{K}^{-1}$ and the heat capacity $C_s$ to $3 \times 10^6\,\text{J}\,\text{m}^{-3}\,\text{K}^{-1}$ (values from thermal model calibration RMSE1; Sect. 6.4).

The results of the six-parameter optimization on field resistivity data are shown in Fig. 8. Figure 8a shows the best fit of apparent resistivities after calibration of the freezing season of 2014/2015. Although the fit between simulated and field apparent resistivities is not ideal, the optimized parameterization produces a temperature field that fits the field temperatures in the freezing season of 2014/2015 within $\pm 0.66\,^\circ\text{C}$ (Fig. 8b).

To assess the predictive value of the model, we used the optimized parameterization estimates (values as listed in Fig. 8a and b) to forward-calculate the apparent resistivity distribution in the freezing seasons of 2012/2013 (Fig. 8c) and 2013/2014, respectively (Fig. 8e). The parameter values optimized on the freezing season of 2014/2015 predicted the field temperature measurements from the freezing season of 2012/2013 with a mean error of $0.38\,^\circ\text{C}$ (Fig. 8d). The field temperature measurements from the freezing season of 2013/2014 (only 2 months are available for comparison) were predicted with a mean error of $0.30\,^\circ\text{C}$ (Fig. 8f).

In terms of the model fit, the performance of the coupled optimization approach is comparable to the traditional optimization on borehole temperatures (Sect. 6.4). In terms of determination of the true parameters of the real ground, the coupled inversion approach does not improve the non-uniqueness of parameter optimization; the optimized parameter values depend on their initial parameter estimates. Nevertheless, all the optimization runs come up with physically plausible parameter values, and the forward-calculated temperature fields fit the field datasets with mean error of around $0.6\,^\circ\text{C}$.

## 8  Discussion

Efforts using geophysical data to constrain other – especially hydrological – models are by now well documented. The coupling strategies vary, from constraining inversion and interpretation of the other models with inverted geophysical data (Doetsch et al., 2013), through structurally coupled approaches (Gallardo and Meju, 2011; Lochbühler et al., 2013), to fully coupled inversion schemes using the geophysical data before inversion (Hinnell et al., 2010; Herckenrath et al., 2013b; Tran et al., 2017). The fully coupled approaches have been encouraged by some (Gallardo and Meju, 2011), as separate data inversions lead to inconsistent models for the same subsurface target. The fully coupled framework has been shown to improve accuracy and reduce the uncertainty of the prediction of hydrological parameters (Hinnell et al., 2010; Herckenrath et al., 2013a).

To distinguish the principle limitations of the presented fully coupled inversion method from possible pathways for improvement, a number of conceptual simplifications and assumptions stated in Sect. 4 are hereby evaluated.

Heat conduction is assumed to be the dominant process of heat transport. This is a reasonable assumption at our site, where for most of the year (from beginning of September to mid-June) we do not observe substantial lateral water movement (Jessen et al., 2014; Tomaškovičová and Ingeman-Nielsen, 2023). Evaporation and increased pore water movement produce water content variations in the unfrozen period (mid-June to end-August), but this is outside of the calibration period (1 September–28 February). Accounting for the processes of water movement and evaporation would require their description and parameterization, which could be handled effectively outside of the coupled inversion scheme. Including such modules would only be of interest if the goal was to carry out year-round ground temperature modeling.

Full saturation is a valid assumption at our site between the beginning of September and mid-June (Tomaškovičová and Ingeman-Nielsen, 2023), and only the data from the fully saturated periods were used in the calibration. It is important to highlight that this is *not* a principle limitation of the method. If the saturation is known, it is straightforward to include its true value in the current model formulation. If we were to optimize for saturation different from 1 but constant in time, then it would correspond to adding just another parameter to the heat conduction (and the coupled) model. However, to include saturation in a realistic way, it should be included as a time-variable parameter (and as a space-variable parameter when expanding the method to more space dimensions). This could be done in one of two ways: by (i) parameterizing saturation (in a way similar to the parameterization of $\theta_w$ by the two parameters $\alpha$ and $\beta$), which would result in additional parameters to calibrate, or by (ii) adding a hydrological model driven by the same climatological parameters that would calculate saturation and inform the coupled thermal–resistivity model about the time-variable values of saturation. Depending on the hydrological model used, it may become necessary to include additional parameters, e.g., describing subsurface hydraulic properties, in the calibration routine.

The assumption of homogeneous ground is to a certain extent a simplification at our site. While the geology on our site is indeed homogeneous in terms of soil type (silty clays, based on geotechnical drilling reported in Geoteknisk Institut, 1978), heterogeneities are present in the form of ice lenses (mainly in the depth between 0.9–1.5 m) and increas-

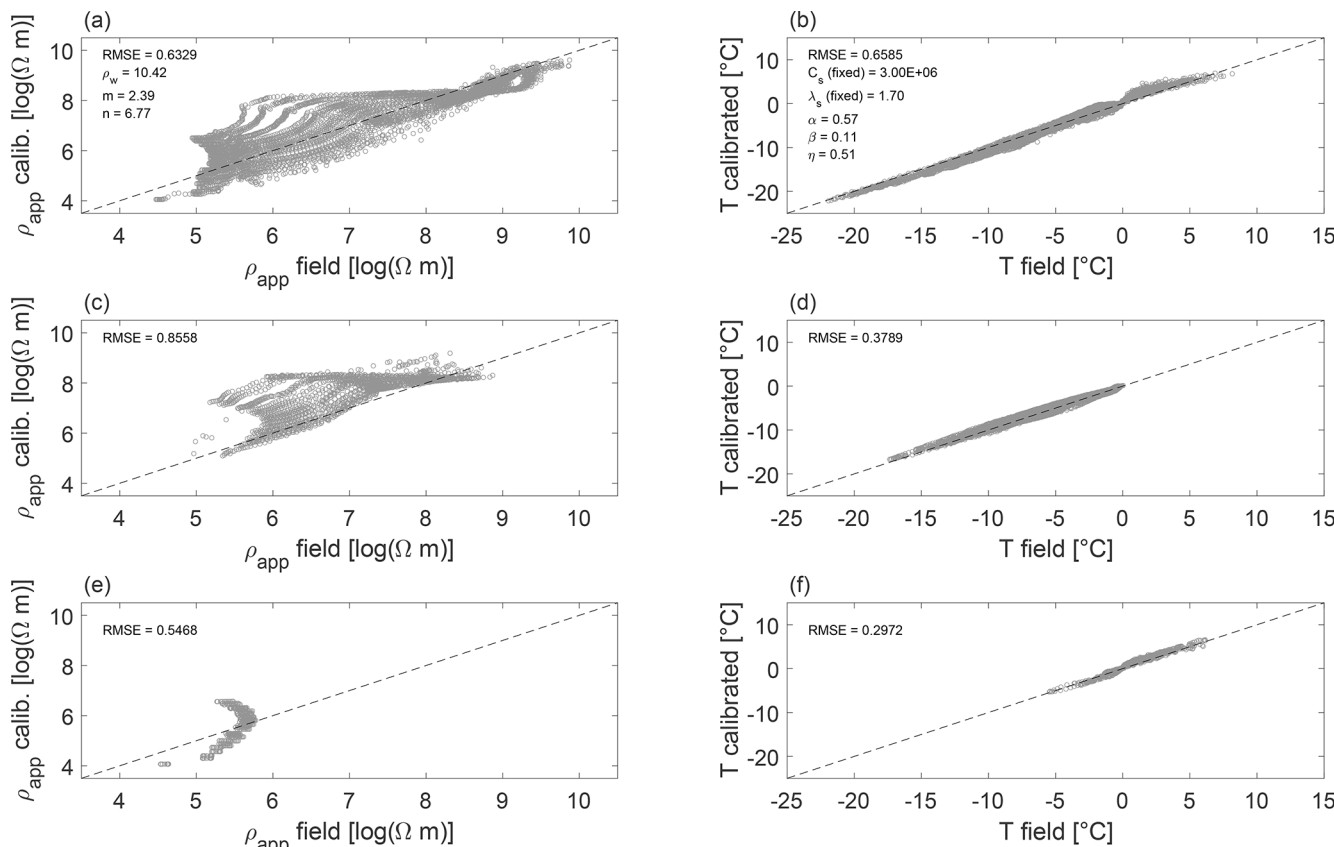

**Figure 8.** Results of the fully coupled inversion: panels **(a)** and **(b)** show results of calibration, and panels **(c)** through **(f)** show results of validation. **(a)** Fit of the simulated apparent resistivities ($\rho_{\text{app}}$) to field $\rho_{\text{app}}$ measured during freezing season of 2014–2015; **(b)** fit of the heat conduction model after optimization, freezing season of 2014–2015. Panels **(c)** and **(d)** show the validation of the optimized coupled model on the freezing season of 2012–2013. Panels **(e)** and **(f)** show the validation of the optimized coupled model on the freezing season of 2013–2014. The values of the fixed and optimized thermal and resistivity model parameters are listed in annotations. RMSE is in degrees Celsius (°C).

ing pore water salinity (in the depth below 4 m). In the homogeneous model setup, the specific thermal properties of soil constituents, the porosity and the freezing curve parameterization are assumed to be the same for the entire soil column. However, the model does resolve varying *bulk* (ef- [5] fective) thermal properties, as these depend on the temperature and phase distribution of the soil constituents at a given time and depth in the soil column. We experimented with an implementation of a heterogeneous model with four lay- [10] ers corresponding to the field situation (active layer, ice-rich permafrost with ice lenses, saline permafrost and bedrock). The optimization algorithm searched for four different sets of the specific thermal parameters, one set for each of the model layers. The performance of the heterogeneous model [15] was comparable to the homogeneous model in terms of the RMSE (°C), and different sets of parameters were identified for each of the four layers. However, the parameters remained non-uniquely determined, as different sets of starting values converged to different optimized parameter values, [20] all fitting the field data similarly well. We concluded that,

without further constraining information, the use of the more complex, heterogeneous model was not justified, in terms of neither the model fit, the speed of convergence nor the uniqueness of parameter estimation. Constraining information could be added, e.g., in the form of orthogonally deter- [25] mined fixed model parameters or a constrain on parameter change (e.g., that porosity has to be decreasing with depth).

Specific thermal properties of the soil constituents were assumed constant, i.e., independent of the temperature or salinity. This is an acceptable approximation, previously used by, [30] e.g., Nicolsky et al. (2007) and Romanovsky et al. (2000). According to Osterkamp (1987), using constant parameters resulted in errors of less than 10 % in the calculation of the effective (bulk) thermal properties in the temperature range between −20–0 °C. Latent heat of phase change in reality [35] varies with unfrozen water content; however, using a constant value has been proved satisfactory for temperatures above −20 °C (Anderson et al., 1973). These are standard assumptions used in modeling experiments designed following the principle of parsimony and where the focus is not on the [40]

temperature and salinity dependence of the thermal parameters. Nevertheless, temperature- or salinity-dependent variation could be readily implemented if the requirements of model output and quality of input data justified the increased model complexity. Including the effect of salinity would additionally require adaptation of the freezing curve formulation.

A fixed temperature was used as the bottom model boundary. This assumption has no impact on our calculations considering the short modeled time span (up to 180 d) and the known yearly temperature amplitude ($< 0.09\,°C$) at the bottom of the 6 m deep borehole (Tomaškovičová and Ingeman-Nielsen, 2023). If the model was to be used for future predictions, this assumption would have to be reviewed, just as in any model expanding its domain of application. Including variable bottom temperature or geothermal heat flux boundaries is technically straightforward in the current model setup. This would, however, require extra data input in the form of borehole temperature measurements (for the variable bottom boundary) or a reasonable estimate of the geothermal heat flux at the site (from a deep borehole or from a regional geothermal heat flux model accounting for variations).

Experiences show that the volumetric heat capacities $C$ of materials are difficult to estimate from typical field and laboratory tests (Shonder and Beck, 1998; Kojima et al., 2018). Our forward temperature field calculations are relatively insensitive to the value chosen for the volumetric heat capacity of soil minerals $C_s$. A plausible explanation for this could be that significant phase change is necessary to separately estimate heat capacity and thermal conductivity from field data; in a system with little phase change, only thermal diffusivity can be estimated. In a saturated system, when the rate of phase change is the largest, the energy consumed by the phase transition between water and ice is much larger than the energy needed to change the temperature of soil grains, and the value of $C_s$ becomes insignificant. Outside of the period of significant phase change, the phase change may not be enough to allow for separate calibration of the heat capacity of the soil grains.

The use of Archie's law is usually limited to sediments with low clay content, when virtually all conductivity in the bulk soil can be attributed to the pore liquid. This condition is not entirely met in our field situation, as high clay content may contribute to lowering the overall soil resistivity by surface conduction. Adaptation of the resistivity–mixing relationship remains a possibility for improvement of the performance of the coupled inversion framework.

Although we advocate for the use of easy-to-measure ground surface temperatures to drive the model, we do recognize that these typically suffer from rapid fluctuations influenced by short-wave radiation. Using daily averages or near-surface temperatures to drive the model instead (at ca. 10 cm depth) could improve performance in the upper portion of the modeled domain. Performance in the deeper parts of the model would be improved by using a heterogeneous model

setup, allowing one to capture vertical variations in the specific soil properties.

The value of replacing borehole temperature data with geophysical data for thermal model calibration could be discussed, as the geoelectrical data achieved calibration comparable to the borehole calibration, though not better. Admittedly, in the absence of directly measured thermal properties, borehole data are the best calibration and validation data for a 1D thermal model. However, depending on the survey requirements and limitations, geophysical data afford a number of benefits that may constitute a practical advantage over the use of borehole temperatures, evidently without sacrificing the model performance:

1. *Measurements from the surface rather than the need for drilling*. This encompasses two advantages: larger depth reach, as well as the possibility to work in both sedimentary and bedrock settings. Hand-operated, engine-powered drilling tools have limited depth penetration and are restricted to sedimentary geology. The logistics associated with mobilizing a drilling rig able to reach greater depths or drilling through bedrock is often prohibitive in remote arctic areas. Meanwhile, the depth reach of a geoelectrical array can be more readily adjusted by the design of the largest spacing of the current electrodes.

2. *Lower impact on fragile ecosystems*. Few roads exist in the Arctic, and the movement of drilling equipment on the tundra, especially outside of the frozen season, seriously damages the terrain, particularly in wetter and ice-rich permafrost areas (Rickard and Brown, 1974; Kevan et al., 1995). Arctic tundra are characterized by relatively low biological activity and diversity and by short, cool and dry growing seasons. This leads to the natural re-vegetation process after surface disruption being very slow. The disruption of the surface organic layer then typically results in the accelerated thaw of permafrost. Together with the risk of pollution from engine-operated equipment, these factors may cause issues securing the necessary permits for drilling fieldwork. In comparison, the impact of the surface or airborne geophysical methods is minimal.

3. *Assessment of spatially varying conditions*. Geophysical mapping methods, unlike point borehole measurements, allow for a relatively quick assessment of ground conditions over a comparatively large areas. Therefore, expanding the presented approach to three-dimensional mapping presents another potential for future development of the method.

Spatial variability of the ground electrical properties is resolved by a combination of electrode layouts on a heterogeneous half space because of the different depth sensitivities of different layouts. This applies to the 1D scenario as well

as to the 2D and 3D scenarios, where different layouts are necessary to cover the part of the subsurface of interest. A real concern is if the equivalencies observed in the inversion of resistivity data from permafrost (Ingeman-Nielsen, 2006; Ingeman-Nielsen et al., 2008; Tomaškovičová and Ingeman-Nielsen, 2023) impact this type of inversion. Such a question is targeting a more complex situation than what we focused on in the simple conditions of this study; more work is necessary to understand how the method performs in more complex settings.

While we illustrated the coupled approach using geoelectrical data, in principle, any geophysical data could be used as long as the petrophysical relationship between the ground temperatures are the geophysical parameter can be calibrated. For example, transient electro-magnetic data (TEM), acquired by towing the instrumentation behind a snowmobile (Kass et al., 2021; Maurya et al., 2021) would exploit the same petrophysical relationship while providing greater spatial coverage. In some settings, TEM acquired from an airplane could be of interest – if the benefit of fast acquisition and great coverage would outweigh the greater cost. These considerations suggest the far-reaching potential of the concept of the coupled thermo–geophysical inversion.

For each new type of geophysical data to be used in the coupled inversion scheme, the required measurement repeat frequency would be of interest. This frequency would depend on the rate of phase change (which varies throughout the year) and the sensitivity of the method to these changes. We observed (Tomaškovičová and Ingeman-Nielsen, 2023) that in the temperature range between $-2$ and $0\,°C$, the unfrozen water content changes very rapidly. Data from this period also contain the most information for the thermal model calibration. Hence, it makes sense to collect geophysical acquisition daily, when possible. When daily acquisitions are not feasible, as well as outside of the period of the fastest phase change, the acquisition frequency can be optimized by first evaluating the effect of sampling frequency on the parameter recovery in a synthetic exercise.

Other types of geophysical data may benefit from the input of independently measured model parameters constraining the model. The identification of such parameters would depend on the sensitivity of the geophysical method of choice and on the desired outcome of the model. Naturally, having a fix on an important parameter such as, e.g., porosity (a single value throughout the soil column or porosity variable with depth) would improve the estimation of the remaining parameters. This would make sense to implement if the purpose of the model was to obtain as close to true parameter values as possible without directly measuring them on soil samples. However, porosity is also one of the trickiest parameters to impose. If the purpose of the modeling is to come up with a model that can reproduce the calibration and validation data within certain acceptable error limits, then this can be achieved even when the true porosity is not known and is one of the calibrated parameters.

The performance of the coupled inversion method at sites of very different geology is of interest for expanding the applicability of the method. While definite answers cannot be given without testing, we can base our assessment on the knowledge of the behavior of the ground thermal parameters, as well as the results of our sensitivity analysis. A site where the ground consists of coarse-grained dry sediment would present a very different scenario to the saturated silty clay geology on which we developed and tested the coupled inversion method. The parameters that we would expect to differ the most in such conditions would be porosity, saturation, and the freezing curve parameterization $\alpha$ and $\beta$. Porosity, $\alpha$ and $\beta$ were among the most sensitive parameters of the thermal model. The sensitivity of the model to saturation was not evaluated in this study; however, as it is a property that directly controls the amount of conductive liquid phase, we would expect it to be a very sensitive parameter. Based on these considerations, we expect that the method would likely be able to resolve the thermal regime even at a site of different geology as long as there is some moisture present in the ground and phase change takes place.

The approach described in this work constitutes one of a number of possible ways of adding constraining information directly to the process of estimation of the heat conduction model parameters. The field geoelectrical data were shown to contain constraining information for the calibration of the thermal model. Even though we did not obtain an ideal resistivity model, the thermal calibration was useful. The fit and predictive performance of the resistivity-calibrated thermal model were comparable to the fit of thermal model calibrated on borehole temperature measurements.

## 9   Conclusions

The two main conclusions of our study can be summarized as follows.

1. We evaluated the amount of information contained in the time lapse geoelectrical data for the calibration of a soil heat conduction model.

2. We demonstrated that the geoelectrical calibration data are useful as alternative calibration data for a heat conduction model and can provide as useful calibration results as borehole temperature data.

This is the first time that field data have been used to demonstrate that the concept of the fully coupled thermo–geophysical inversion can work in praxis. The fully coupled model achieved a performance comparable to the traditional method of calibration of borehole temperatures. While the model did not necessarily improve the estimation of thermal parameters compared to the calibration of borehole temperature measurements, it provided an alternative way of deriving it from surface measurements. Additionally, the thorough

sensitivity analysis of the model parameters improved our understanding of what geological information may be necessary for constraining the model.

In the development process of the fully coupled inversion scheme, we thoroughly evaluated a comparatively simple (1D, homogeneous, three-phase) model for heat transfer in a ground undergoing cycles of freezing and thawing. The model relies on a number of conceptual assumptions to maintain parsimony. Nevertheless, it predicts temperature variation at our test site with satisfactory accuracy – within $0.55\,°C$. The simplicity of the model is a benefit in that the requirements of input data are relatively low – only surface temperature time series (measured or downscaled from air temperature data), initial temperature distribution and bottom boundary condition are needed. On the other hand, it has to be expected that the final, optimized parameter estimates will compensate for conceptual simplifications of the model.

In the context of geotechnical and engineering applications (such as forecasting stability of infrastructure built on thawing permafrost), the true values of thermal parameters remain of interest, as they can be used further in geotechnical models. Further efforts in improving the structure and sensitivity of the model, constraining the optimization, and including further independent validation will be required to validate the model in this context.

The relatively short time series of one freezing season (180 d) were sufficient to reach a plausible parameter estimation for the freezing season, providing good fit to the validation dataset. Due to hysteretic nature of freeze–thaw processes, the heat conduction model should be calibrated for freezing and thawing seasons separately.

With the constantly improving understanding of electrical resistivity responses in the very specific permafrost settings, the resistivity coupling with thermal models opens up new possibilities for monitoring the current and forecasting the future thermal state of permafrost. The method has the demonstrated potential for offering a more resource-efficient alternative to calibration of ground thermal models on borehole temperature records.

*Code availability.* Soilfreeze1D – the 1D thermal conduction forward calculation code – is available here: https://github.com/tingeman/soilfreeze1d (Ingeman-Nielsen, 2023a).

CR1Dmod – the resistivity forward modeling code – is available here: https://github.com/tingeman/CR1DMOD (Ingeman-Nielsen, 2023b).

The inversion code makes use of the cited Matlab optimization toolbox functions; the exact setup is available upon request from the corresponding author.

*Data availability.* The underlying research data are available upon request from the corresponding author.

*Author contributions.* ST conducted the data treatment and modeling and wrote the paper. TIN implemented the soilfreeze1D solver for heat conduction and contributed to the intellectual content of the paper. Both authors contributed to development of the coupled inversion scheme.

*Competing interests.* The contact author has declared that none of the authors has any competing interests.

ther geographical representation in this paper. While Copernicus Publications makes every effort to include appropriate place names, the final responsibility lies with the authors.

*Acknowledgements.* This study was part of project Tarajulik, funded by the Greenland Research Council (grant agreement no. 80.30). The study was also supported by the Nunataryuk project, which is funded under the European Union's Horizon 2020 Research and Innovation Programme (grant agreement no. 773421).

*Financial support.* This research has been supported by Horizon 2020 (grant no. 773421) and the Nunatsinni Ilisimatusarnermik Siunnersuisoqatigiit (grant no. 80.30).

*Review statement.* This paper was edited by Christian Hauck and reviewed by two anonymous referees.

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

**Remarks from the typesetter**