# Peer review of "Coupled thermo-geophysical inversion for permafrost monitoring"

_The Cryosphere, 2023_

## Author Comment (AC1)

Author's response

to reviewers on the initial submission of the preprint tc-2023-51

"Coupled thermo-geophysical inversion for permafrost monitoring"

by Soňa Tomaškovičová & Thomas Ingeman-Nielsen

1. June 2023

>> We thank the reviewer for the thoughtful review and for acknowledging the value of the presented work. We are particularly thankful for the excellent suggestions on how to improve the discussion section of the paper. All of the reviewer's suggestions were thoroughly considered and implemented in the revised manuscript, which we look forward to uploading upon the editor's invitation.

Our replies are indicated in ">> the blue paragraphs".

**Reviewer 1**

The manuscript "Coupled thermo-geophysical inversion for permafrost monitoring" by Tomaškovičová and Ingeman-Nielsen presents a new inversion algorithm for ERT measurements in permafrost which uses a heat conduction model to constrain the possible subsurface temperature and thus resistivity distributions. Although the presented method still has serious limitations, it is an important step towards a more quantitative use of ERT measurements for permafrost monitoring. The manuscript is well-written and follows a logical structure. I have a few comments which the authors should resolve before the manuscript can be published.

-The authors should consider using a more standard structure with for example Introduction, Methods (Sects.2-5), Results (Sects. 6-7), etc. In my opinion, this will improve the organization of the manuscript.
>> We have previously considered a more standard article structure. The current structure highlights the methodological aspect of developing and validating a coupled inversion framework by sectioning the text into the respective model development steps. We consider this to be one of the main contributions of the paper and would prefer to keep the current structure, if possible.

-L. 81: I suggest replacing «advice» by «yield»

>> Edited in the revised version of the manuscript.

-L. 144: This is a very restrictive assumption which severely limits the usability of the method. At the study site (i.e. where the ERT system is deployed), is the soil really fully saturated? In general, these limitations of the method should be evaluated in more detail in the Discussion section, also clarifying if they are principle limitations, or if they may be overcome by future work. See comment below!

>> At the study site in Greenland, the full saturation is a good assumption, certainly for most of the year except between mid-June and end of August (based on consistent in-situ unfrozen water content measurements over three years). Also, it is only the fully saturated period that is used for calibration in the presented work, so the assumption is not violated.

For the thermal model, as long as the saturation is known, it is straightforward to include it in the formulation. If the saturation was unknown but constant in time, then optimizing for saturation would correspond to adding just another parameter to the heat conduction (and the coupled) model.

So, the current restriction of the method is that the saturation has to be considered constant in time.

To include saturation in a realistic way, it should be included as a time-variable parameter. This could be done in one of two ways: by i) parameterizing saturation (in a way similar to the parameterization of *theta_w* by the two parameters *alpha* and *beta*), which would of course add additional parameters to calibrate; or by ii) adding a hydrological model driven by the same climatological parameters that would calculate saturation and inform the coupled thermal-resistivity model about the time-variable values of saturation. This way, there would be no additional parameters in the coupled model to calibrate.

-Sect. 6 and its subsections: I am not sure "heat model" is a good term, the authors could consider using "heat conduction model" instead.

>> We agree to use "heat conduction model"; alternatively, we have introduced the shorter expression "thermal model" where brevity was needed.

-Sect. 6.1: The reasoning of this section is not too clear for me, as the authors first seem to treat all model parameters equal, not making use of the knowledge that some are essentially constants, while others are not constrained at all. Why do the authors consider the heat capacities of water/ice and thermal conductivities of water/ice as potentially variable parameters? Why not use the accepted literature values for these, potentially including parametrizations for their temperature dependence? What is the expected freezing point at the study site, is it close to 0? Also the volumetric heat capacity of soil minerals should be in a fairly small range, other than the thermal conductivity which might vary by a factor of three? So why not use a literature value for the heat capacity, is the idea to account for the effect of soil organics? In the end (Table

2), exactly the parameters that are generally assumed constant in environmental modeling, are assumed constant also here, using pretty much the literature values (on a side note, why use Tf=-0.0001 degrees and not 0 - this shouldn't affect the model outcome at all?). So why not restrict the sensitivity analysis to the parameters which in reality are variable, and drop the constant ones from the sensitivity analysis?

>> We found it interesting to demonstrate the sensitivity of the model to all of its parameters (as per the heat conduction equation (1)), including the ones that are in practice not calibrated (constants), to showcase their individual importance for model simulations, for two main reasons:

1) Such constants would normally be excluded from a sensitivity analysis, while we found it interesting to show the effect of changing values of these parameters on the model calculation.
2) These 'constants' are, strictly speaking, not constant, but temperature- and salinity-dependent (specific heat capacity and thermal conductivity of ice and pore water). The assumption of a constant is a standard simplifying modeling assumption that we found justifiable in a study *not* focusing on the temperature- and salinity dependence of these thermal parameters, and designed following the principle of parsimony. Nevertheless, a known temperature and salinity dependence could easily be incorporated into the current model formulation, if the model application required it and justified the increased complexity.

-L. 243: it would be good to specify "arbitrary, but realistic" some more. Is it "randomly drawn within the limits of Table 2"?

>> Indeed, Table 2 presents a wide range of values found across literature, and the "arbitrary but realistic" values are the most common values we found for our type of materials, from the values within this table.

-L. 251: Write out MRC. Is this mentioned before?

>> MRC is a rigid thermistor string, we have replaced the abbreviation in the revised version of the manuscript.

-L. 284: "of the permitted…"?

>> Edited in the revised version of the manuscript.

-Sect. 6.4: It would be good to include a discussion of the physical meaning of the Cs-parameter and if that can help understand the relative insensitivity of modeled temperatures towards this parameter.

>> The relative insensitivity of the model to the Cs (heat capacity of soil grains) parameter is well-known from experimental and modeling studies, but we found the causes to be little discussed. Our reasoning departs from the knowledge that significant

phase change is necessary to separately estimate heat capacity and thermal conductivity from field data; in a system with little phase change, only thermal diffusivity can be estimated. In a saturated system, when the rate of phase change is the largest, the energy consumed by the phase transition between water and ice is much larger than the energy needed to change the temperature of soil grains, and the value of Cs becomes comparatively insignificant. Outside of the period of signification phase change, the phase change actually taking place is not enough to allow for separate calibration of the heat capacity of the soil grains. This discussion was included in the revised version of the manuscript.

-L. 297/Table 2: use unit MJ instead of e6

>> Edited in the revised version of the manuscript.

-L. 310/311: Can you rephrase this sentence, I'm not sure what starting model-dependent means.

>> It means that the final, optimized parameter values will be different based on the initial guess of the parameter value before optimization. Rephrased in the revised version of the manuscript.

-Fig. 5 and Fig. 8: add unit to RMSE

>> Edited in the revised version of the manuscript.

-L. 398: in Fig. 8

>> Edited in the revised version of the manuscript.

-Sect. 8 Discussion: In my opinion, the discussion is missing a detailed assessment of the state of the proposed method, how it would be used in practice, i.e. what type of ERT measurements one would need (e.g. repeat frequency), which soil parameters need to be measured independently to constrain the model, etc.

>> The repeat frequency would depend on the rate of phase change (which varies throughout the year) and the sensitivity of the geophysical method to these changes. During the fastest rate of phase change (notably in the thawing period), it is beneficial to collect geophysical data daily, when possible. When daily acquisitions are not feasible, as well as outside of the period of the fastest phase change, the acquisition frequency could be optimized by evaluating the effect of sampling frequency on the parameter recovery. A discussion point on the repeat frequency was added to the revised manuscript.

In terms of independent model parameters to be measured to constrain the model - these would depend on the sensitivity of another geophysical method and on the desired outcome of the model and would need to be tested.

Naturally, having a fix on an important parameter such as e.g. porosity (a single value throughout the soil column, or variable porosity with depth) would improve the estimation of the remaining parameters. This would make sense to do if the purpose of the model was to obtain as close to true parameter values as possible (without directly measuring them on soil samples). However, porosity is also one of the trickiest parameters to know, and impose. If the purpose of the modeling is to come up with a model that can reproduce the calibration and validation data well (within acceptable errors), this can be achieved even when porosity is not known and is one of the calibrated parameters.

Secondly, it would be nice to discuss to what extent the performance of the method at the Greenland site is specific to this site, or if at least a roughly similar performance can be expected also at other sites, where e.g. some of the ground parameters could be different (). While this is clearly a difficult point and definite answers cannot be given, the authors could use the results of their various sensitivity analyses together with general knowledge on ground parameters (e.g. thermal conductivity) and the ranges within which they vary.

>> Definite answers cannot be given without testing, but we can base our assessment on the knowledge of the behavior of the ground thermal parameters, as well as the results of our sensitivity analysis. A site where the ground consists of coarse-grained dry sediment would present a very different scenario to the saturated silty clay geology on which we developed and tested the coupled inversion method. The parameters that we would expect to differ the most would be porosity, saturation, and the freezing parameterization alpha and beta. Porosity, alpha and beta were among the most sensitive parameters of the thermal model. Saturation sensitivity was not evaluated in this study, yet as it is a property that directly controls the amount of conductive liquid phase, we would expect it to be a very sensitive parameter. Based on these considerations, we expect that the method would likely be useful at a site of different geology. Discussion on how we expect the method would work at a site of different geology was added to the revised manuscript.

The authors should also discuss the impact of the limitations stated in L.141 on the practical use of the method and if it is possible to improve/adapt the method to overcome at least some of these limitations.

>> *Heat conduction assumption*: Heat conduction is assumed to be the dominant process of heat transport. This is a reasonable assumption at our site where for most of the year (from beginning of September to mid-June), we do not observe substantial lateral water movement. Evaporation and increased pore water movement produce water content variations in the unfrozen period (mid-June to end-August), but this is outside of the calibration period (1. September -- 28. February). Accounting for the processes of water movement and evaporation would require their description and parameterization, which could be handled effectively outside of the coupled inversion

scheme. Including such modules would only be of interest if the goal was to carry out year-round ground temperature modeling

>> *Full saturation assumption:* discussed in a reply to an earlier comment.

>> *Homogeneous ground assumption:* The assumption of homogeneous ground is indeed a simplification at our site. Heterogeneities are present, namely in the form of ice lenses (mainly in the depth between 0.9 - 1.5 m), and increasing pore water salinity (in the depth below4m). We implemented a heterogeneous model with four layers corresponding to the field situation (active layer, ice-rich permafrost with ice lenses, saline permafrost and bedrock). The performance of the heterogeneous model was comparable to the homogeneous model in terms of the RMSE, and different sets of parameters were identified for each of the four layers. However, the parameters were non-uniquely determined, as different sets of starting values converged to different optimized parameter values, all fitting the field data similarly well. We concluded that without further constraining information, the use of the more complex model was not justified, neither in terms of the model fit, speed of convergence, or the uniqueness of parameter estimation.

>> *Constant specific thermal properties and latent heat:* Specific thermal properties of the soil constituents were assumed constant, i.e. independent of the temperature or salinity. This is an acceptable approximation, as using constant parameters resulted in errors of less than 10% in the calculation of the effective (bulk) thermal properties in the temperature range between-20 - 0 degrees C. Latent heat of phase change varies with unfrozen water content, but using a constant value has been proved satisfactory for temperatures above \SI{-20}{\celsius} \citep{anderson1973unfrozen}. These are standard assumptions used in this type of modeling, and were previously successfully used in other (cited) studies. Temperature- or salinity-dependent variation could be readily implemented if the requirements on model output and quality of input data justified the increased model complexity.

>> *Fixed temperature as the bottom boundary*: A fixed temperature was used as the bottom model boundary. This assumption has no impact on our calculations considering the short modeled time span (up to 180 days), and the known yearly temperature amplitude at the bottom of a 6m deep borehole (<0.09 degree). If the model was to be used for future predictions, this assumption would have to be reviewed, just as in any model expanding its domain of application. Including variable bottom temperature or geothermal heat flux boundaries is technically straightforward in the current model setup. This would however require extra data input in the form of borehole temperature measurements (for the variable bottom boundary) or a reasonable estimate of the geothermal heat flux at the site (from a deep borehole, or from a regional geothermal heat flux model accounting for variations).

This discussion of all six model assumptions was added to the revised version of the manuscript.

---

## Author Comment (AC2)

Author's response

to reviewers on the initial submission of the preprint tc-2023-51

**"Coupled thermo-geophysical inversion for permafrost monitoring",**

by Soňa Tomaškovičová & Thomas Ingeman-Nielsen

1. June 2023

>> We are thankful to the reviewer for the thoughtful review and for acknowledging the value of the presented work. We are particularly thankful for the excellent suggestions on how to improve the discussion section of the paper. Our replies are indicated in ">> the blue paragraphs". All of the reviewer's suggestions were thoroughly considered and implemented in the revised manuscript, which we look forward to uploading upon the editor's invitation.

**Reviewer 2**

This paper describes the development and application of a fully-coupled inversion to retrieve thermal soil properties from geophysical and temperature data. The paper describes in detail how the models and inversion are setup, what their assumptions and limitations are, and by doing so provides a great reference on how to develop such a fully-coupled approach. The authors provide a detailed sensitivity study, showing that the most critical parameters for the estimation of subsurface thermal properties are the porosity and the soil grains thermal properties.

Their synthetic and field study shows that the soils thermal parameters can be estimated reasonably well using the temperature data, even though the soil is modelled as homogeneous layer. Adding the resistivity data is successful, but does not seem to contribute to improve the parameter estimates.

>> Indeed, it is true that in our study, the resistivity data does not provide a calibration superior to the calibration on borehole temperatures. This is because the intention of this study was 1) to evaluate the amount of information in geophysical data for calibration of a soil thermal model, 2) show that geophysical calibration data are useful as alternative calibration data and provide as good results as borehole temperature data. In certain situations, the use of geophysical data may have practical advantages that we discuss further in reply to the reviewer's comment below on "3. Value of geophysical data".

Although the paper presents the developed approach very clearly, and provides a significant amount of detail on the approachs performance, I'm missing a more detailed discussion on the limitations of the approach, which I would summarize as follows:

41 Modelling of a 1D distribution of thermal properties: While you state that your synthetic tests did not justify using a more complex thermal field, your discussion should address this issue. While it is well known that the thermal properties of the subsurface do vary with depth, why is the model not able to resolve this? Is it because you would require stronger temperature gradients, i.e. are the sensitivities to small to able to resolve this variability? And if resolving of vertical variations is not possible, how do your results improve current understanding of soil thermal property variations?

>> We distinguish between the bulk (effective) thermal properties of the ground vs. the specific thermal properties of the ground constituents ( of water, ice, soil grains).

The model *is* able to resolve the effective thermal properties of the ground as they vary with depth; these properties are effectively changing with depth and with temperature in the modeled domain. The current implementation of the model can also handle prior constraining information in terms of known thermal parameters and known geological boundaries.

What the model - in the presented implementation - doesn't optimize for, is different *specific* thermal properties of the respective soil constituents (eg. thermal conductivity and heat capacity of the soil matrix) that could be varying with depth if the geology varied. The reason for this is that the geology on our site is, based on geotechnical boreholes, homogeneous in terms of soil type (silty clays). Therefore it is justified to optimize for uniform specific soil thermal properties throughout the soil column. (but the bulk/effective thermal properties are still different, as they depend on temperature and phase distribution of the soil constituents in the soil column). It would be interesting to explore how e.g. inverting for varying porosity through depth would perform, but constraining information may need to be added (e.g. porosity has to be decreasing with depth). This would be an interesting question for a follow-up study.

Our model is not necessarily aiming at improving the current understanding of soil thermal properties variations, but at providing an alternative way of deriving it from surface measurements. Showing the sensitivity of parameters improves the understanding of what geological information may be necessary for constraining the model.

We edit the related statements in the revised version of the manuscript.

51 Spatial variability: While you already discuss that there is a lack of model performance in the unfrozen state where water flow may contribute to temperature variations, you neglect the spatial variability of the electrical properties. In line 63 you state: "The relationship translating a certain ground electrical composition into apparent resistivity is unique", while this is true in the

way you state it, it is not true for the inverse, and thus provides a major limitation for your inversion that is not discussed. I.e. a homogenous subsurface distribution may provide the exact same apparent resistivity response than an arbitrarily layered medium and this will become even more complex when going from 1D to 3D. While this may not be a major limitation in some geological settings, in permafrost environments where the electrical properties are highly heterogenous, I would argue that this is a major limitation of your approach.

>> This concern is valid for a single geoelectrical measurement, or for several measurements on a homogeneous half-space, but not for a combination of electrode layouts on a heterogeneous half-space because of the different depth sensitivity of different layouts. This applies also to 2D and 3D scenarios, where different layouts are necessary to cover the part of the subsurface of interest.

A real concern is if the equivalencies observed in the inversion of resistivity data from permafrost impact this type of inversion. This is currently not known. Such a question is targeting a more complex situation than what we focused on in the simple conditions of this study. More work is necessary to understand how the method performs in more complex settings.

We've added this point to the Discussion in the revised manuscript.

61 Value of the geophysical data: You describe in much detail the performance of the thermal parameter inversion, which seems to provide very good results. Yet, when you add the geophysical data, the performance seems to degrade (i.e. you need to fix Cs to obtain reasonable estimates) and hence I am wondering what the rational is to include the resistivity data. Clearly, it would allow you to assess spatially varying parameter distributions, but this is not shown here.

>> We would like to stress that this study presents the results of an experimental phase where we investigated to which extent we can replace (not supplement) the borehole temperature data with geophysical data. Attempts at exploiting information in geophysical data that can be interpreted in terms of temperatures are also known from the Alps, where geophysical surveys are used to construct virtual boreholes. This study is another such approach to trying to exploit geophysical data in terms of their information content about ground temperatures.

In some situations, borehole temperature data may be the best calibration data. However in certain conditions, the following practical advantages of geophysics could be of interest:

1) Measurements collected from the surface rather than the need for drilling: This encompasses two advantages: larger depth reach, as well as the possibility to work in both sedimentary and bedrock settings. Hand-operated, engine-powered drilling tools are of limited depth penetration and restricted to sedimentary geology. Logistics associated with mobilizing a drilling rig able to reach larger depths or drilling through bedrock is often prohibitive in remote arctic areas. Meanwhile, the depth reach of a geoelectrical array can be more readily adjusted by the design of the largest spacing of the current electrodes.

2)  Smaller impact on fragile ecosystems: Few roads exist in the Arctic, and the movement of drilling equipment on the tundra, especially outside of the frozen season, seriously damages the terrain, particularly in wetter and ice-rich permafrost areas. Arctic tundras are characterized by relatively low biological activity and diversity, and by short, cool and dry growing seasons. This leads to the natural re-vegetation process after surface disruption being very slow. The disruption of the surface organic layer then typically results in the accelerated thaw of permafrost. Together with the risk of pollution from engine-operated equipment, these factors may cause issues securing the necessary permits for drilling fieldwork. In comparison, the impact of the surface or airborne geophysical methods is minimal.

3)  Assessment of spatially varying conditions: Geophysical mapping methods, unlike point borehole measurements, allow for a relatively quick assessment of ground conditions over comparatively large areas. Therefore, expanding the presented approach to three-dimensional mapping presents another potential for future development of the method.

We have added these points to the Discussion section of the revised manuscript.

In summary, while I think that the paper very nicely presents the develop approach, and the thermal inversion seems to provide reasonable results, it remains unclear what the benefit of including the geophysical data really is and how the limiting assumptions really affect the model outcome. I think that needs to be stated much more clearly, and will require a more detailed discussion section.

>> We thank the reviewer for acknowledging the presentation of the method. A discussion of the benefit of using the geophysical data has been added in reply to the previous comment (and added to the revised Discussion of the manuscript). Additionally, we think that this study, while presenting a working method, also has its value as a proof of concept and an exploration of possible pathways for future development.

Below are some more specific remarks:

Lines 20 - 21: I find this misleading as it seems like there are only 3 studies that look into petrophysical relationships, but there are a number of papers, e.g. Olhoeft (1978), Magnin et al. (2015), Hoekstra and McNeil (1973), Scott and Kay (1988), Holloway and Lewkowicz (2019), Tang et al. (2018), Uhlemann et al. (2021), Wu et al. (2017)

>> We had no intention of making it look like only three studies were available. More relevant references were listed in the discussion. We have reviewed the suggested references and added them to the revised version of the manuscript.

Lines 54-56: Would this require you to know the composition of the ground? If so, how would you obtain that information, which likely is spatially varying too.

>> We appreciate the reviewer's positive attitude about expanding the method into more dimensions. The purpose of this study is not to investigate 2-3D variability, but rather whether resistivity contains information to invert for the thermal properties, and whether

it can replace borehole temperatures in 1D setting. The resistivity method already has its limitations and in 2D, the resistivity inversion relies on constraining info to resolve 2D variations (smoothness constraints). We are of the opinion that first, we needed to find out if the approach was possible in 1D before expanding into more dimensions. With the successful results reported here, it would be interesting to expand the analysis to 2D and 3D. It is conceivable that in such cases, there may be a need to combine resistivity data with other constraining or calibration information (other geophysics, remote sensing).

"Composition of the ground" - does it refer to different soil types? Or different proportions of ground constituents (water, ice, soil grains)? The latter is solved by the model. Regarding different soil types, it would be of course very interesting to test the performance of the method at a site of different geology; for this, at least one freezing season of resistivity data from a different site would be needed.

Lines 146 - 149: Wouldn't this have two explanations: (1) you are not sensitive to these variations, or (2) the uncertainty of your results are larger than the vertical variability?

>> We think this is addressed in the reply to the reviewer's comment above "1.Modelling of a 1D distribution of thermal properties". Also, the statements are clarified in the revised version of the manuscript and discussion on the topic is added.

Figure 2: Here and elsewhere (also in the text), you first refer to the thermal conductivity as k, but then change its annotation to lambda.

>> Notation changed to lambda throughout the revised version of the manuscript.

References:

Olhoeft, G. R. (1978). Electrical properties of permafrost. Proceedings of the Third International Conference on Permafrost, 127–131.

Magnin, F., Krautblatter, M., Deline, P., Ravanel, L., Malet, E., & Bevington, A. (2015). Determination of warm, sensitive permafrost areas in near-vertical rockwalls and evaluation of distributed models by electrical resistivity tomography. Journal of Geophysical Research, 120, 745–762. https://doi.org/10.1002/2014JF003351

Hoekstra, P., & McNeill, D. (1973). Electromagnetic probing of permafrost. North AmericanContribution to the Second International Conference on Permafrost, 517–526.

Scott, W. J., & Kay, A. E. (1988). Earth Resistivities of Canadian Soils.

Holloway, J. E., & Lewkowicz, A. (2019). Field and laboratory investigation of electrical resistivity-temperature relationships, southern Northwest Territories. Cold Regions Engineering, 64–72. https://doi.org/10.4324/9780203210536

Tang, L., Wang, K., Jin, L., Yang, G., Jia, H., & Taoum, A. (2018). A resistivity model for testing unfrozen water content of frozen soil. Cold Regions Science and Technology, 153, 55–63. https://doi.org/10.1016/j.coldregions.2018.05.003_chapter_9

Uhlemann, S., Dafflon, B., Peterson, J., Ulrich, C., Shirley, I., Michail, S., & Hubbard, S. S. (2021). Geophysical monitoring shows that spatial heterogeneity in thermohydrological dynamics reshapes a transitional permafrost system. Geophysical Research Letters, 48(6), 1–11.

https://doi.org/10.1029/2020gl091149

Wu, Y., Nakagawa, S., Kneafsey, T. J., Dafflon, B., & Hubbard, S. (2017). Electrical and seismic response of saline permafrost soil during freeze - thaw transition. Journal of Applied Geophysics, 146, 16–26. https://doi.org/10.1016/j.jap

---

## Author Response (AR2)

*From:* *Soňa Tomaškovičová (corresponding author), Thomas Ingeman-Nielsen*

*To:* *Editorial Office, The Cryosphere*

In Kongens Lyngby, DK, 03. August 2023

**Subject**: Response to the second round of reviews. Upload of 2nd revision (3rd version) of the manuscript.

Dear Editor and Reviewers,

Thank you for your suggestions on how to improve our submission of the manuscript "*Coupled thermo-geophysical inversion for permafrost monitoring*". We are hereby submitting the 2nd revision of the manuscript, where we implemented the second round of changes. In addition to replies to reviewers' comments listed on the following pages, we have:

- rephrased the Abstract for conciseness and clarity.
- replaced the reference to PhD thesis (Tomaškovičová 2018) with a reference to a just published article (Tomaškovičová & Ingeman-Nielsen, 2023) "*Quantification of freeze–thaw hysteresis of unfrozen water content and electrical resistivity from time lapse measurements in the active layer and permafrost*" (doi: https://doi.org/10.1002/ppp.2201). This changed/updated reference was highlighted in red and used whenever we refer to the field site description, to the thermal, resistivity or water content regime at the site, and to the challenges with inverting resistivity acquisitions from permafrost settings.

Changes in the manuscript (compared to the 1st revision) are marked with red text.

We hope that you will find our updated contribution suitable for publication in The Cryosphere.

Respectfully,

Soňa Tomaškovičová

Department of Environmental &
Resource Engineering
Section for Geotechnics & Geology
Technical University of Denmark

Nordvej 119
2800 Kgs. Lyngby
Denmark

Tel.. +45 45 25 50 98
Mail: soto@dtu.dk

www.sustain.dtu.dk

[Figure]

**Response to reviewers**

*>> We thank the reviewers for repeatedly reviewing the manuscript and for providing further comments to help us improve it. Our replies are indicated in ">>italics".*

**Reviewer 1**

Dear authors,

thank you very much for addressing my and the other reviewer's comments and for providing detailed responses to our suggestions. The extended discussion really benefits the paper, and it is much clearer now why you chose to use the resistivity, and also what the current limitations are and how they may be addressed in the future. While I think that the paper can be accepted now, I just want to come back to one of my previous comments.

In lines 64 to 67 you state:

"The relationship translating a certain ground electrical composition into apparent resistivity is unique, and governed by equations for conservation of charge, Ohm's law and the geometry of electrode configuration used to collect the resistivity data. Conversely, any inverted resistivity model is only one of a large number of possible realizations that explains the measured apparent resistivity data acceptably well"

While I fully agree that working with apparent resistivities is beneficial (e.g., avoiding inversion constraints, dealing with non-uniqueness, etc.), I still find these two sentences misleading. The way you state the first sentence is of course correct. You can use the results of your thermal model to create a 1D subsurface representation from which you can calculate a unique resistivity response. But as for the inversion of apparent resistivity, where an infinite number of subsurface models will be able to explain your measured apparent resistivities, an infinite number of thermal parameters will be able to explain your measured apparent resistivities. With these two sentences, for someone who is not familiar with resistivity measurements, it sounds though as if there is a unique solution to your optimization problem. This then also feeds into the problem of spatial heterogeneity. In your model, you solve for a 1D subsurface, with no spatial only vertical variation in geophysical and thermal properties. And even if your set of apparent resistivity measurements will be sensitive to spatial heterogeneity, you are not addressing this in your model (or at least I don't see how). This may not be a problem at your field site, but, e.g., when working on polygonal ground, where subsurface parameters can vary at small spatial scales, it would certainly affect your apparent resistivity measurements, and thus may lead to errors in your coupled inversion. Of course, this cannot be addressed in a 1D model and is certainly outside the scope of this paper, but it might be something to keep in mind.

*-> We have added a clarifying statement to make this more obvious, in lines 72-75.*

**Department of Environmental &**
**Resource Engineering**
**Section for Geotechnics & Geology**
Technical University of Denmark

Nordvej 119
2800 Kgs. Lyngby
Denmark

Tel.. +45 45 25 50 98
Mail: soto@dtu.dk

www.sustain.dtu.dk

[Figure]

**Reviewer 2**

In general, the authors have provided convincing answers to my comments. However, the only major change to the manuscript is the much extended discussion, while the rest is more or less unchanged. Wherever applicable, the authors should incorporate the replies in the other parts of the manuscript as well (at least if the point is not taken up in the revised discussion, in this case please provide a reference to the discussion section where the original comment was raised). An example is my comment on Sect. 6.1. I have no problem with the explanations provided in the replies, but when reading the manuscript without these explanations in mind, I still have the same problem to understand the logic behind the setup. So please incorporate the replies in the manuscript!

*>> We have tried to incorporate the points from the discussion throughout the manuscript where relevant:*

- *in the expanded description of the heat model assumptions (section 4, lines 151-171)*
- *in section 6.1, lines 220-227.*

Minor points:

L.208: use the greek symbol for lambda, not "lambda"

*>> Corrected.*

Table 1: Please explain for each of the parameter how the value (or the range) was selected.

*>> Explanation added in lines 261-265.*

L. 423: I don't understand how saturation could be "parameterized" (point i) if it changes in time. What would be the input parameters for this parameterization, how can the soil water content respond to precipitation, evaporation, etc.

*>> Parameterization would require knowledge of the soil moisture regime on the site. In our case, in the frozen period, for T < Tf: S=1. In the thawed period, saturation dependence would have to be investigated, e.g. it is known respond to two main forcing regimes: rainfall-driven wetting or radiation-driven drying.*

L. 425: I don't agree that no additional parameter would need to be optimized when a hydrological model is added. In fact, a hydrological model would likely have multiple unknown parameters which strongly influence the soil water content and which would need to be optimized in the inversion unless they are known from other studies. Examples would be the hydraulic conductivity (which e.g. determines how fast water infiltrates vertically and drains laterally), the water holding capacity of the soil or soil hydraulic parameters (if for example Brooks-Corey or Mulaem-van Genuchten models for matric potentials are used).

**Department of Environmental &**    Nordvej 119      Tel.. +45 45 25 50 98      www.sustain.dtu.dk
**Resource Engineering**    2800 Kgs. Lyngby      Mail: soto@dtu.dk
**Section for Geotechnics & Geology**    Denmark
Technical University of Denmark

[Figure]

*-> We completely agree that the hydrological model would need its own parameterization. The idea is that no additional parameter would be optimized within the coupled thermo-geophysical inversion framework, i.e. the hydrological model would be a separately calibrated model driven by climate variables. This would obviously greatly increase demands on input data amount and quality, and would only be justified if adequate data quality and hydrological model were available, and if a part of the year with variable saturation was used.*

L. 447: If the soil is saline, the entire model for soil freezing needs to be adapted to account for the freezing point depression and changes to the soil freeze curve. This should be clarified. The fact that the soil thermal parameters do not vary strongly with salinity is a minor point compared to this.

*>> Completely agree. At the same time, the focus of this paragraph is justifying the use of constant specific thermal parameters, rather than explaining how salinity should be incorporated. We have however included a sentence about the type of change necessary if the salinity was to be included (line 485-486).*

L. 475: I don't understand this point, the daily temperature fluctuations are real, so they shouldn't influence the results of the algorithm. Is this mainly a computation problem, i.e. the model has a longer computation time if it needs to resolve the daily temperature fluctuations? In this case, it should be no problem to use daily averages of the ground surface temperature to drive the model. Or is it hard to resolve the effects of the surface layer which can have very different properties from the ground below (e.g. organic surface layers)?

*>> According to Figure 5, the thermal model has shown to smoothen the temperature variation compared to the in-situ measurements. Use of daily averages could be a solution, however it should be tested how their use impacts the modeling results.*

**Department of Environmental &**       Nordvej 119             Tel.. +45 45 25 50 98           www.sustain.dtu.dk
**Resource Engineering**                2800 Kgs. Lyngby        Mail: soto@dtu.dk
**Section for Geotechnics & Geology**   Denmark
Technical University of Denmark

---

## Author Response (AR3)

**From:** *Soňa Tomaškovičová (corresponding author), Thomas Ingeman-Nielsen*

**To:** *Editorial Office, The Cryosphere*

In Kongens Lyngby, DK, 06. October 2023

**Subject**: Response to the third round of reviews. Upload of the production files.

Dear Editor and Reviewers,

Thank you for reviewing our manuscript *"Coupled thermo-geophysical inversion for permafrost monitoring"*. We are hereby submitting the files for production, where we have implemented the suggested change:

*In summary, the statement "This way, there would be no additional parameters in the coupled model to calibrate" is at least not true in its generality, and I suggest replacing it by: "Depending on the hydrological model used, it may become necessary to include additional parameters, e.g. describing subsurface hydraulic properties, in the calibration routine."*

The changed statement appears in the lines 459-461 of the revised manuscript (version for production).

Respectfully,

*Soňa Tomaškovičová*

Soňa Tomaškovičová

**Department of Environmental &**      Nordvej 119            Tel.. +45 45 25 50 98           www.sustain.dtu.dk
**Resource Engineering**                2800 Kgs. Lyngby       Mail: soto@dtu.dk
**Section for Geotechnics & Geology**  Denmark
Technical University of Denmark